# Disease-modifying effects of sodium selenate in a model of drug-resistant, temporal lobe epilepsy

Pablo M Casillas-Espinosa[1,2,3]*, Alison Anderson[1,2], Anna Harutyunyan[1,2], Crystal Li[2], Jiyoon Lee[1], Emma L Braine[1,2], Rhys D Brady[1,2], Mujun Sun[2], Cheng Huang[4], Christopher K Barlow[4], Anup D Shah[4], Ralf B Schittenhelm[4], Richelle Mychasiuk[2], Nigel C Jones[1,2], Sandy R Shultz[1,2], Terence J O'Brien[1,2,3]*

[1]Department of Medicine, The Royal Melbourne Hospital, The University of Melbourne, Melbourne, Australia; [2]Department of Neuroscience, Central Clinical School, Monash University, Melbourne, Australia; [3]Monash Proteomics & Metabolomics Facility and Monash Biomedicine Discovery Institute, Monash University, Clayton, Victoria, Australia; [4]Department of Neurology, The Alfred Hospital, Commercial Road,, Melbourne, Victoria, Australia

*For correspondence:
pablo.casillas-espinosa@monash.edu (PMC-E);
Terence.OBrien@monash.edu (TJO'B)

Competing interest: The authors declare that no competing interests exist.

**Abstract** There are no pharmacological disease-modifying treatments with an enduring effect to mitigate the seizures and comorbidities of established chronic temporal lobe epilepsy (TLE). This study aimed to evaluate for disease modifying effects of sodium selenate treatment in the chronically epileptic rat post-status epilepticus (SE) model of drug-resistant TLE. Wistar rats underwent kainic acid-induced SE or sham. Ten-weeks post-SE, animals received sodium selenate, levetiracetam, or vehicle subcutaneousinfusion continuously for 4 weeks. To evaluate the effects of the treatments, one week of continuous video-EEG was acquired before, during, and 4, 8 weeks post-treatment, followed by behavioral tests. Targeted and untargeted proteomics and metabolomics were performed on post-mortem brain tissue to identify potential pathways associated with modified disease outcomes. Telomere length was investigated as a novel surrogate marker of epilepsy disease severity in our current study. The results showed that sodium selenate treatment was associated with mitigation of measures of disease severity at 8 weeks post-treatment cessation; reducing the number of spontaneous seizures ($p < 0.05$), cognitive dysfunction ($p < 0.05$), and sensorimotor deficits ($p < 0.01$). Moreover, selenate treatment was associated with increased protein phosphatase 2A (PP2A) expression, reduced hyperphosphorylated tau, and reversed telomere length shortening ($p < 0.05$). Network medicine integration of multi-omics/pre-clinical outcomes identified protein-metabolite modules positively correlated with TLE. Our results provide evidence that treatment with sodium selenate results in a sustained disease-modifying effect in chronically epileptic rats in the post-KA SE model of TLE, including improved comorbid learning and memory deficits.

## Editor's evaluation

This important study provided evidence that sodium selenate is a treatment that reduces the symptoms of chronic seizures in a rat model of epilepsy. The improved symptoms were reduced seizure frequency, improved memory, and improved sensorimotor function. The authors provided convincing evidence that even after treatment ended the benefits persisted, a remarkable finding. They also identified several possible mediators of the effects, including PPAR and h-tau, as well as telomere length. An in-depth study of potential genetic regulation by selenate provided additional possible targets of selenate that ameliorate epilepsy.

**eLife digest** According to the World Health Organization (WHO), there are around 50 million people with epilepsy worldwide. Although drugs are available to control epileptic seizures, these only provide symptomatic relief. They cannot prevent the condition from worsening, and if people with epilepsy stop taking their medication, there is no lasting effect on the severity or frequency of their seizures.

Some epilepsy cases are also resistant to these drugs. This is particularly common in adults with temporal epilepsy, with 30% of people continuing to suffer with seizures despite receiving medication. Current treatments also have no effect on problems with learning, memory and mental health that sometimes accompany drug-resistant epilepsy.

Previous studies in animals have identified some potential treatments that could slow the progression of temporal epilepsy, but these have only been shown to work when used at a very early stage. Since most individuals with temporal epilepsy have already started having seizures when they are diagnosed (and it is difficult to predict who will develop the condition), these drugs are unlikely to be useful in practice.

Here, Casillas-Espinosa et al. set out to find if a novel drug called sodium selenate can stop the progression of epilepsy and reduce the severity of temporal epilepsy when the condition is fully advanced. To do this, they used an animal model of temporal epilepsy, where rats had been modified to develop spontaneous seizures, resistance to normal anti-seizure medications, and problems with learning and memory.

Casillas-Espinosa et al. found that sodium selenate not only reduced the number and severity of seizures in these model rats, but also improved their memory and learning ability. Several rats stopped having seizures altogether even after the treatment had stopped, indicating that sodium selenate had a long-lasting protective effect. Genetic analysis of the rats also revealed that shorter telomeres (special DNA sequences at the ends of chromosomes) correlated with increasing severity of the condition, suggesting that telomere length could help predict who might develop temporal epilepsy or respond best to treatment.

This study identifies sodium selenate as a potential treatment that could reverse the progression of temporal epilepsy, even in individuals with advanced symptoms. Later this year, sodium selenate will be trialled in people with drug-resistant temporal epilepsy to determine if the drug benefits humans in the same way. Casillas-Espinosa et al. hope that it will improve participants' epilepsy and, ultimately, their quality of life.

## Introduction

Temporal lobe epilepsy (TLE) is the most common form of drug-resistant epilepsy in adults and is frequently accompanied by disabling neuropsychiatric and cognitive comorbities (*Hermann et al., 2008*; *Hermann et al., 2000*; *Kwan et al., 2011*; *Sharma et al., 2007*; *Tellez-Zenteno et al., 2007*). People with epilepsy are often required to take multiple anti-seizure medications (ASMs) in an attempt to control their seizures. However, these treatments are ineffective at controlling seizures in more than 30% of cases (*Hermann et al., 2008*; *Hermann et al., 2000*; *Kwan et al., 2011*; *Sharma et al., 2007*; *Tellez-Zenteno et al., 2007*), and can cause significant adverse side effects (*Alonso-Vanegas et al., 2013*; *Perucca and Gilliam, 2012*; *Taylor et al., 2011*). Current ASMs are merely symptomatic and do not mitigate the severity, progression, or outcome of the epilepsy, and if the patient ceases the medication their seizures are just as severe and frequent as if the person has never taken the medication in the first place (*Simonato et al., 2014*). Despite decades of research and development, the more than 20 new ASMs that have been introduced to clinical practice have failed to reduce the proportion of people with epilepsy who have drug-resistant seizures (*Chen et al., 2018*). Moreover, the cognitive and neuropsychiatric comorbities that are prevalent in these patients remain difficult to treat and are not improved by the ASM treatment (*Hinnell et al., 2010*; *Valente and Busatto Filho, 2013*).

Animal studies have identified compounds potentially capable of delaying epileptogenesis (i.e. the process by which a non-epileptic brain turns into an epileptic one), but to date, these have only been demonstrated to be effective when the treatment is initiated early in the disease process (i.e. immediately after the brain insult) (*Casillas-Espinosa et al., 2019b*; *Liu et al., 2016*; *Pauletti et al.,*

*2019*; *Rizzi et al., 2019*; *Zeng et al., 2009*; *Huang et al., 2019*). Because the majority of TLE patients already have established epilepsy when they present to the clinic (*Foster et al., 2019*), and there are no reliable biomarkers to predict who will develop TLE, this treatment strategy has limited clinical utility (*Simonato et al., 2021*). Surgical resection of the epileptogenic zone is currently the only effective disease-modifying intervention in patients with chronic drug-resistant TLE, but is only suitable in selected patients, is not always successful, and can have significant complications (*Téllez-Zenteno et al., 2005*). It is estimated that <5% of patients with drug-resistant epilepsy are suitable for resective epilepsy surgery (*Téllez-Zenteno et al., 2005*). Thus, there is a compelling need to identify pharmacological DMTs that are effective in patients with chronic TLE. (*O'Brien et al., 2013*; *Sharma et al., 2007*; *Simonato et al., 2013*).

Hyperphosphorylated tau (h-tau) is a neuropathological hallmark of several neurodegenerative conditions and has recently been implicated in epileptogenesis (*Casillas-Espinosa et al., 2020*; *Ittner and Götz, 2011*; *Johnstone et al., 2015*; *Liu et al., 2016*; *Sen et al., 2007*; *Shultz et al., 2015*; *Thom et al., 2011*; *van Eersel et al., 2010*). For example, h-tau is present in the brains of rodents with TLE, (*Corcoran et al., 2010b*; *Jones et al., 2012*; *Shultz et al., 2015*) ,and deposits of h-tau have been identified in surgically resected tissue from patients with drug-resistant epilepsy (*Puvenna et al., 2016*; *Sánchez et al., 2018*; *Sen et al., 2007*; *Tai et al., 2016*; *Thom et al., 2011*). The primary tau phosphatase in the brain (i.e. an enzyme that dephosphorylates tau) is protein phosphatase 2 A (PP2A) (*Corcoran et al., 2010a*), a heterotrimer consisting of a catalytic C subunit, a scaffold A subunit, and a regulatory B subunit with variable isoforms (*Janssens and Goris, 2001*). In several models of epilepsy, activation of PP2A with sodium selenate has been shown to reduce h-tau, and early treatment imparts anti-epileptogenic effects, while also reducing behavioral deficits associated with the epilepsy (*Liu et al., 2016*). However, whether pharmacological intervention with sodium selenate in chronic epileptic animals is a DMT remains to be determined. Therefore, this study investigated the effect of sodium selenate treatment on epileptic and behavioral outcomes, as well as multiomics network medicine and bioinformatics approaches to uncover relevant molecular pathways, in chronically epileptic rats. We used the well-validated kainic acid (KA) post- post-status epilepticus (SE) model of TLE, where animals develop resistance to drug treatment, behavioral, cognitive, and sensorimotor comorbidities analogous to human drug-resistant TLE (*Casillas-Espinosa et al., 2019b*; *Thomson et al., 2020*).

## Results

This study evaluated the effect of sodium selenate on established TLE, using the post-SE model of drug-resistant TLE. Ten-weeks post-SE, chronically epileptic rats were randomly assigned to receive either sodium selenate (1 mg/kg/day), levetiracetam (200 /mg/kg/day, a commonly prescribed ASM for TLE) (*Mbizvo et al., 2012*), or vehicle (saline 0.9%) continuously for four weeks. To evaluate the sustained effects of the treatments, video-EEG was acquired before, during, and up to eight weeks post-treatment, followed by behavioral tests to assess cognitive, and sensorimotor comorbidities. Targeted and untargeted proteomics and metabolomics were performed on brain tissue to investigate potential pathways and molecules associated with modified disease outcomes. Telomere length was investigated as a surrogate marker of disease severity and treatment response. The experimental paradigm is shown in Figure 7 of the Methods section.

### Sodium selenate is disease-modifying with enduring effects that reduce the number of seizures in drug-resistant TLE rats

Video-electroencephalogram (vEEG) was used to examine the effects of the treatments before, during, and up to eight weeks post-treatment on the occurrence of spontaneous recurrent seizures (*Figure 1*). Vehicle and levetiracetam-treated animals showed a significant progressive increase in the number of seizures recorded over the course of the experiments [$F_{(8, 124)}=2.022$, $p<0.05$]. vEEG analysis during continuous drug treatment (10–14 weeks post-SE) revealed that neither vehicle, sodium selenate, or levetiracetam significantly impacted the number [$F_{(4, 62)}=0.4781$, $p=0.751$], duration [$F_{(2, 43)}=0.5030$, $p=0.608$], or severity of the seizures [$F_{(2, 43)}=0.4563$, $p=0.636$], experienced by post-SE rats, indicating that drug-resistant epilepsy is present 10 weeks after KA-induced SE (*Figure 1A*). However, following the cessation of treatment, one-way ANOVA showed a treatment

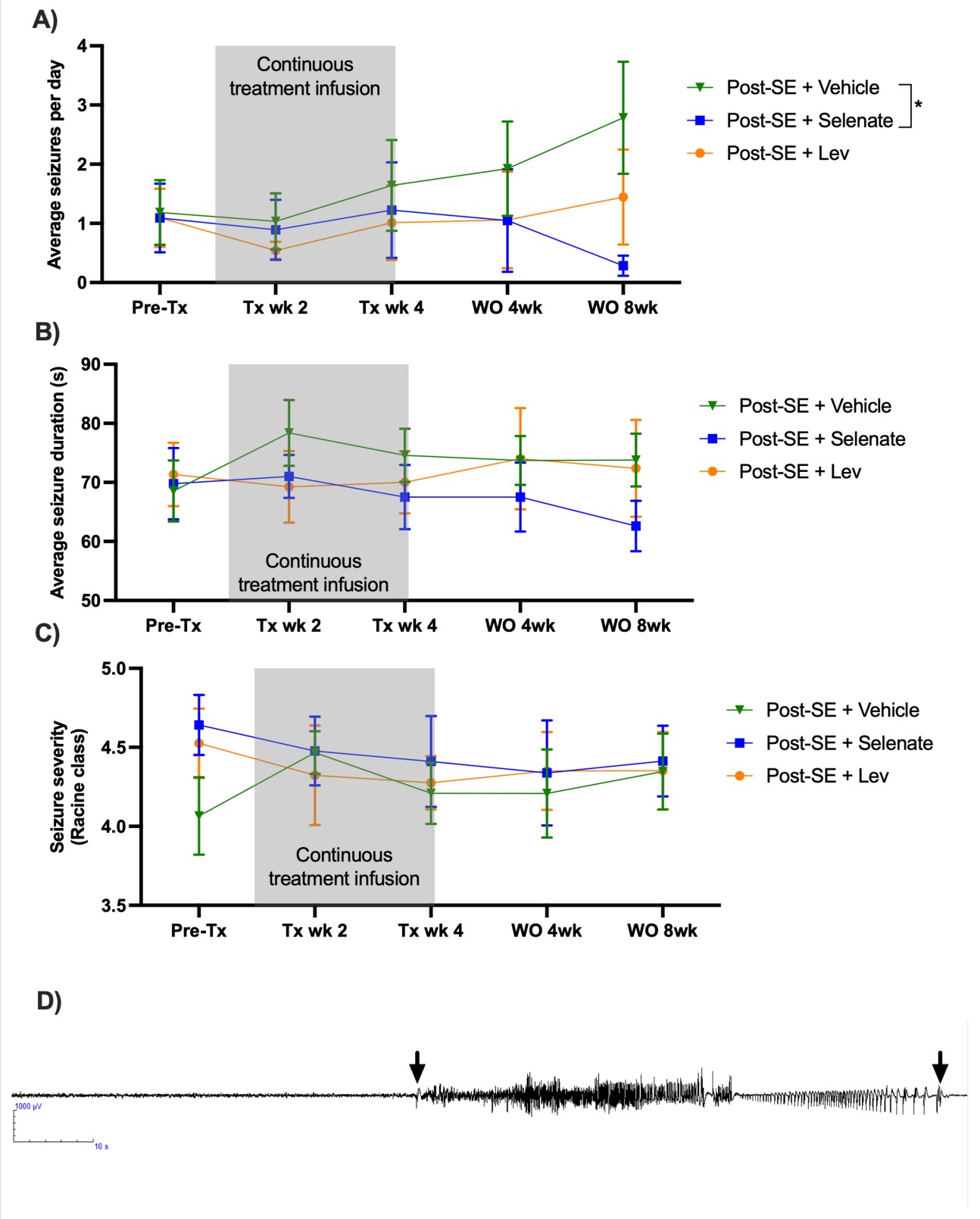

**Figure 1.** Sodium selenate has a sustained effect to reduce spontaneous seizures in the post-KA SE rat model of chronic drug-resistant temporal lobe epilepsy (TLE). (**A**) Sodium selenate stopped the progression of seizures from treatment week four and significantly reduced seizure frequency eight weeks after the last day of treatment compared to vehicle-treated animals (**B**) average seizure duration and (**C**) seizure severity was not significantly different between the three treatment groups; (* significantly different from vehicle p<0.05). LEV levetiracetam. Two-way ANOVA with Dunn's *post hoc*.

*Figure 1 continued on next page*

*Figure 1 continued*

Tx- treatment, wk week, WO, washout. Data shown as mean ± SEM. Post-SE +Vehicle (n=10), Post-SE +selenate (n=12), and Post-SE +levetiracetam (n=12). (D) Representative EEG example of an animal presenting a spontaneous seizure, 120 s window, low pass 70 Hz, high pass 1 Hz; arrows represent seizure start and seizure ends.

The online version of this article includes the following figure supplement(s) for figure 1:

**Figure supplement 1.** Seizure analysis - Single data points.

effect reducing the number of seizures that the animal experienced [$F_{(31, 31)}$=3.956, p<0.0001; *Figure 1A*]. Tukey's post hoc analysis revealed that sodium selenate treatment significantly reduced the number of seizures eight weeks post-treatment in contrast to vehicle-treated animals (p<0.05). Interestingly, four of the twelve sodium selenate-treated animals did not manifest any seizures eight weeks after the drug washout, while all of the vehicle and levetiracetam-treated animals showed spontaneous seizures through the experiment. In contrast, Tukey's post hoc showed that levetiracetam treatment did not significantly reduce the number of seizures in comparison to vehicle or selenate-treated animals. Average seizure duration [$F_{(1, 15)}$=1.038, p=0.3243; *Figure 1B*] and seizure severity, assessed by the Racine scale (*Racine, 1972*) [$F_{(1, 13)}$=0.4873, p=0.497; *Figure 1C*] were not significantly different between the three treatment groups. Figures showing the individual timepoints of the seizure analysis are found in *Figure 1—figure supplement 1*.

## Sodium selenate treatment improves spatial memory and recognition memory

Cognitive deficits are one of the most debilitating comorbidities seen in patients with TLE (*Hermann et al., 2008*; *Hermann et al., 2000*; *Kwan et al., 2011*; *Sharma et al., 2007*; *Tellez-Zenteno et al., 2007*). The novel object placement (NOP) evaluates spatial memory (*Spanswick and Sutherland, 2010*). Vehicle-treated animals had impaired spatial memory in comparison to shams, as they spent significantly less time in the novel location of the object (p<0.05, *Figure 2A*). Levetiracetam treatment did not improve spatial memory compared to vehicles. Animals that had been treated with sodium selenate, even after an eight week washout, spent significantly more time evaluating the novel location of the object than those treated with the vehicle, which is indicative of improved spatial memory (p<0.05). The novel object recognition (NOR) evaluates recognition memory (*Spanswick and Sutherland, 2010*). Both vehicle and levetiracetam-treated rats spent less time evaluating the novel object relative to sham rats (p<0.05, *Figure 2B*), which is indicative of impaired recognition memory. In contrast, sodium selenate treatment was also able to improve recognition memory in comparison to vehicle-treated rats (p<0.05). Interestingly, the performance of the sodium selenate-treated animals was comparable to shams in both the NOP and NOR. Figures showing the individual timepoints of the MWM are found in *Figure 2—figure supplement 1*.

To further evaluate cognitive function, rats were tested in the Morris water maze (MWM) (*Morris, 1984*). During the acquisition session of the MWM, only the post-SE rats treated with vehicle and levetiracetam displayed significantly increased search times than their sham counterparts [subject x treatment effects: $F_{(12, 164)}$=2.501, p<0.01; selenate animals p>0.05 *Figure 2C*]. One day after the water maze acquisition session, rats underwent MWM reversal session where the location of the escape platform was changed. Here, all of the post-SE rats cohorts took more time to find the hidden platform in comparison to shams [$F_{(4, 157)}$=5.971 p<0.001 *Figure 2D*]. No differences in swim speed were observed between the different groups.

## Sodium selenate treatment improves balance in drug-resistant TLE rats

Rats were tested on the beam task to assess balance and coordination as a measure of sensorimotor function, which is commonly affected in people with drug-resistant TLE (*Carter et al., 1999*; *Carter et al., 2001*; *Liu et al., 2016*). Post-SE rats treated with sodium selenate displayed improved sensorimotor function compared to the vehicle as indicated by significantly fewer slips and falls (p<0.01, *Figure 2E*). In contrast, both vehicle and levetiracetam-treated animals showed significantly more slips and falls compared to shams (p<0.001 and p<0.05, respectively). The time to traverse the beam was not significantly different between the groups. No significant findings were observed in either the

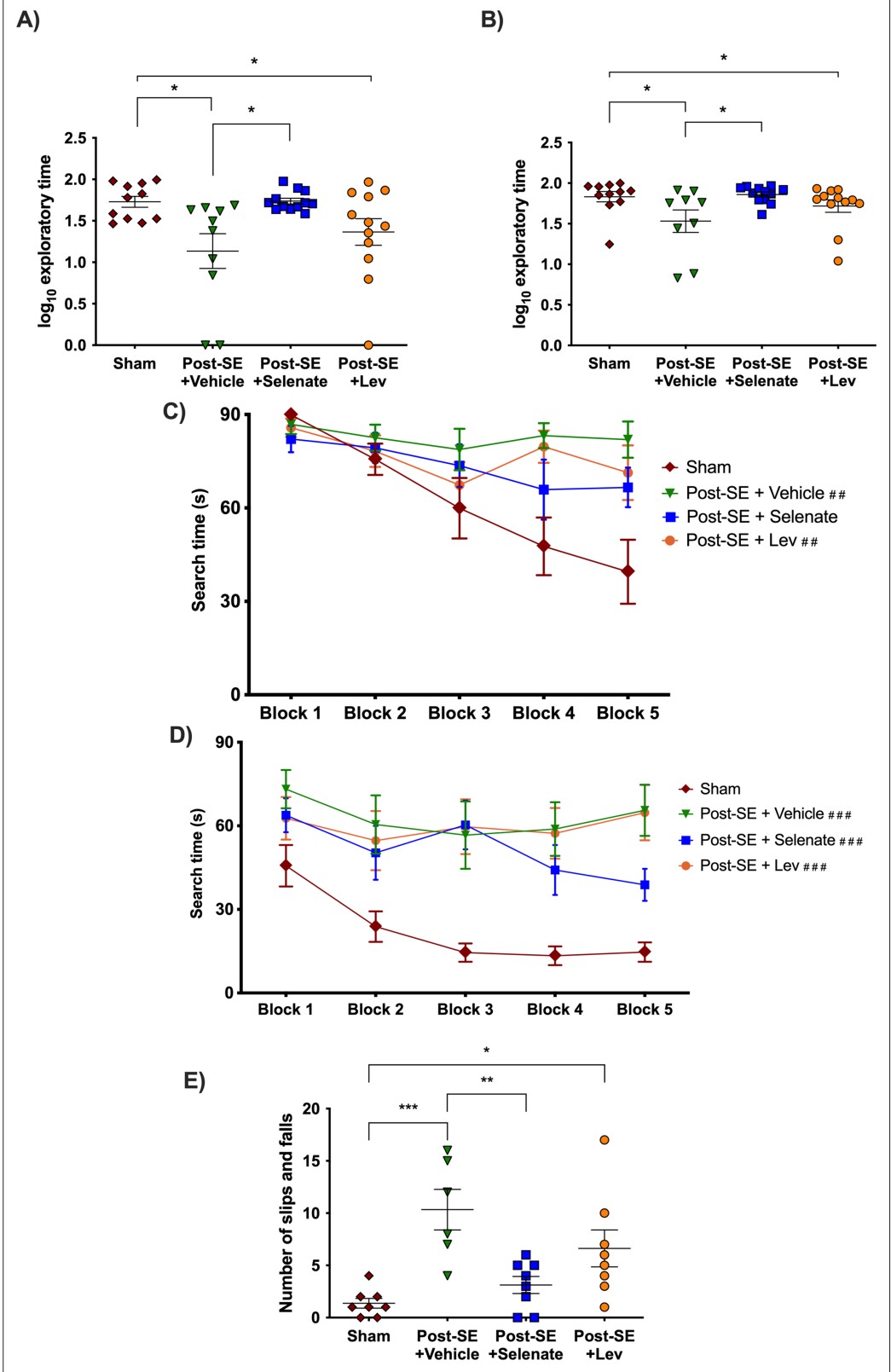

**Figure 2.** Sodium selenate improves cognitive and sensorimotor impairments in the post-KA SE rat model of chronic drug-resistant temporal lobe epilepsy (TLE). (**A**) Novel object placement (NOP). Vehicle and levetiracetam-treated animals showed a significantly reduced time examining the novel location of the object compared to shams. In contrast, sodium selenate treatment improved spatial memory in comparison to the vehicle in post-

*Figure 2 continued*

SE rats. (**B**) Novel object recognition (NOR). Post-SE animals treated with vehicle or levetiracetam showed significantly impaired recognition memory to shams. Sodium selenate rats spent more time evaluating the novel object introduced into the arena in contrast to the vehicle, which is a measure of improved recognition memory. (**C**) Post-SE rats treated with vehicle and levetiracetam had significantly longer search time in the water maze acquisition session in contrast to shams. (**D**) Water maze reversal, all of the post-SE rats showed significantly longer search times in comparison to the sham. For presentation purposes, data for water maze search time is graphed in blocks of two trials. (**E**) Post-SE rats treated with vehicle and levetiracetam exhibited more slips and falls on the beast compared to shams. In contrast, treatment with sodium selenate significantly reduced these impairments compared to the vehicle. (*$p<0.05$, **$p<0.01$, ***$p<0.001$; ##$p<0.05$, and ###$p<0.001$ significantly different to shams). Kruskal-Wallis with Dunn's post hoc for the NOP and NOR. MWM two-way repeated measures ANOVA, with Dunnett's post hoc. One-way ANOVA with Tukey's post hoc for the beam task. Data are shown as mean ± SEM.

The online version of this article includes the following figure supplement(s) for figure 2:

**Figure supplement 1.** Morris Water Maze - Single data points.

**Figure supplement 2.** Post-SE rats show depressive-like behavior as assessed by the sucrose preference test.

open field or elevated plus maze tests of anxiety-like behavior. The results of the sucrose preference test are shown in *Figure 2—figure supplement 2*.

## The protein phosphatase family is differentially expressed after treatment with sodium selenate

Sodium selenate has been described to activate PP2A subunit B and to reduce h-tau after traumatic brain injury and in different models of epilepsy (*Liu et al., 2016*; *Shultz et al., 2015*). However, to further understand the mechanism by which sodium selenate exerts its disease-modifying effects in our model of established chronic TLE, we utilized state-of-the-art, high-resolution data-independent acquisition mass-spectrometry (DIA-MS), untargeted proteomics and metabolomics analyzes were carried out in the somatosensory cortex and hippocampus to identify potential markers of TLE and on the proteome and metabolome. A total of 5615 and 5538 proteins have been identified and quantified across all samples in the somatosensory cortex and hippocampus, respectively, considering a false discovery rate (q-value) cut-off of 1%. Differential expression analyses using ANOVA revealed 24 proteins in the somatosensory cortex and 9 proteins in the hippocampus to be significantly dysregulated across all experimental conditions (*Figure 3*, *Supplementary file 1* and *Supplementary file 2*). Notably, protein phosphatase 2 regulatory subunit B alpha (Ppp2r5a), heat shock protein family A member 2 (Hspa2), and lymphocyte cytosolic protein 1 (Lcp1) were differentially expressed between selenate and vehicle-treated post-SE groups in the somatosensory cortex (Tukey's HSD, FDR <0.05, *Figure 3A*, *Supplementary file 1*). Heat shock protein family B member 1 (Hspb1), Ro690 Y RNA binding protein (Ro60), and DENN domain containing 11 (Dennd11) proteins were differentially expressed between selenate and vehicle treated groups in the hippocampus (Tukey's HSD, FDR <0.05, *Figure 3B*, *Supplementary file 2*). To further examine the changes in the proteomic profile of selenate-treated rats we performed quantitative gene set enrichment analysis between all pairs of treatment groups. *Figure 3—figure supplement 1* shows the top 25 differentially regulated pathways in the somatosensory cortex (*Figure 3—figure supplement 1A*) and hippocampus (*Figure 3—figure supplement 1B*). Interestingly in the somatosensory cortex of the selenate group, among the most significantly upregulated pathways were signaling cascades mediated by activated insulin-like growth factor receptors (IGFR) and fibroblast growth factor receptors (FGRF) such as the FGFR-mediated phosphatidylinositol 3-kinase (PI-3K) cascade activation and IGFR1 signaling cascades (FDR <0.05). Among the top downregulated pathways in the somatosensory cortex were beta-oxidation of fatty acids and iron uptake and transport.

In addition to proteomic analyses, we also performed untargeted univariate metabolomic analyses on the same samples. However, we were not able to identify significant changes between the experimental groups in the somatosensory cortex, suggesting that selenate primarily induces proteomic alterations in this region whilst keeping the metabolome stable (*Supplementary file 3*). In contrast, six metabolites were significantly different in selenate-treated animals as shown in *Supplementary file*

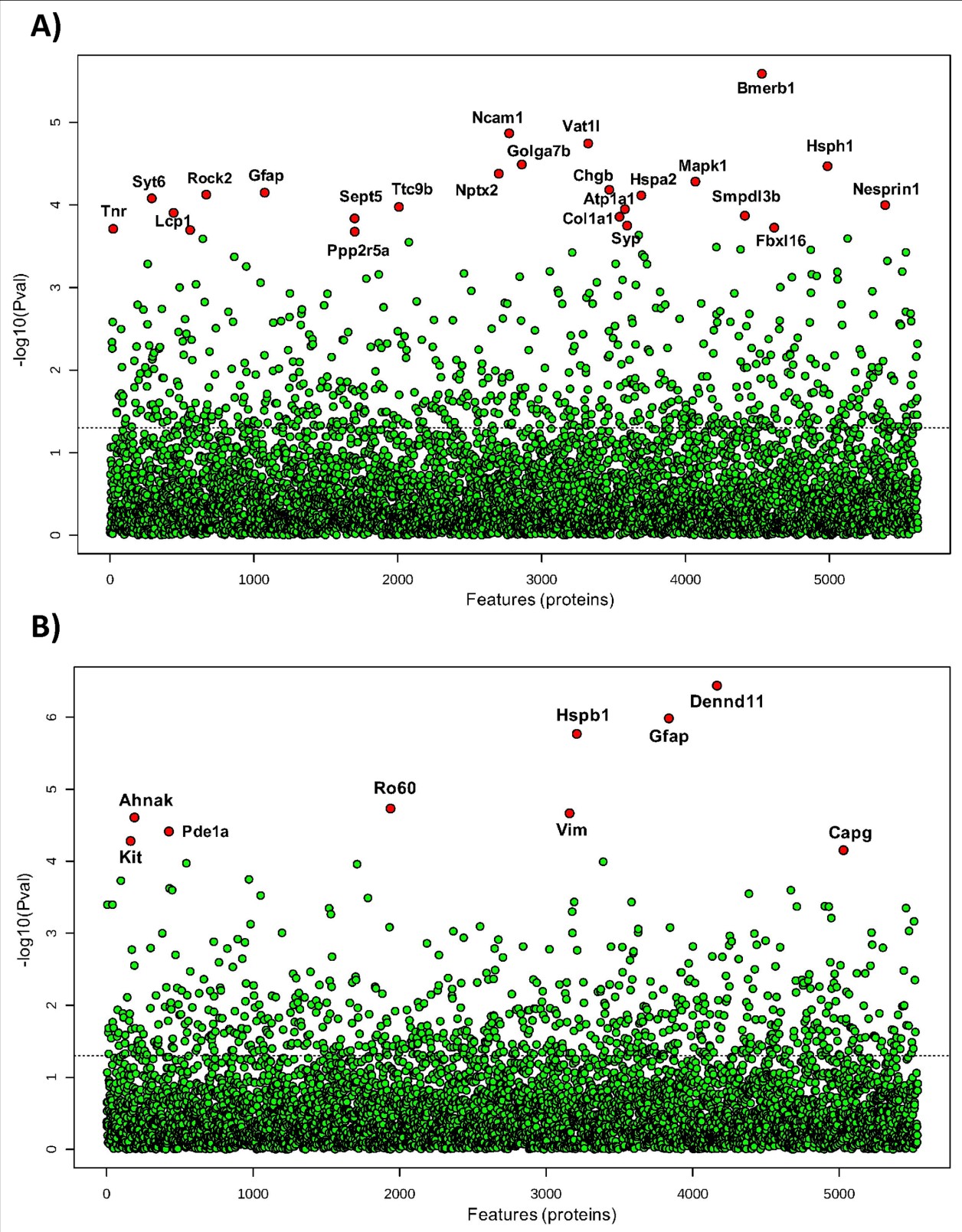

**Figure 3.** *Differential expression analysis of all detected proteins* in (**A**) somatosensory cortex and (**B**) hippocampus of all post-SE treatment groups and shams. One-way ANOVA was used to identify the difference in protein abundance among selenate, levetiracetam, and vehicle-treated groups and shams. Difference between specific groups was determined via Tukey's honestly significant difference (HSD) test. Significance threshold was set to FDR <0.05, and significantly different proteins are labeled red.

*Figure 3 continued on next page*

*Figure 3 continued*

The online version of this article includes the following figure supplement(s) for figure 3:

**Figure supplement 1.** The top 25 differentially regulated pathways in the (**A**) somatosensory cortex and (**B**) hippocampus of selenate treated rats compared to vehicle group.

**Figure supplement 2.** Partial least squares-discriminant analysis (PLS-DA) of the (**A**) proteomics from the somatosensory cortex, (**B**) proteomics from the hippocampus, (**C**) metabolomics from the somatosensory cortex, (**D**) metabolomics from the hippocampus.

**Figure supplement 3.** Sodium selenate modifies hyperphosphorylated-tau and protein phosphatase 2A (PP2A) expression.

---

*4*. Partial least squares-discriminant analysis (PLS-DA) and principal component analysis (PCA) for the proteomics and metabolomics analyses are shown in *Figure 3—figure supplement 2*.

## Sodium selenate treatment persistently reduces h-tau and increases PP2A protein expression

To further validate the modification of the h-tau pathway by sodium selenate in established TLE, we performed targeted protein analysis of h-tau and PP2A in the somatosensory cortex only due to tissue availability. Semi-automated western blots showed that sodium selenate treatment persistently reduced h-tau compared to vehicle and levetiracetam-treated animals (p<0.01 for both comparisons, *Figure 3—figure supplement 3A*). Similarly, PP2A protein expression was significantly increased in selenate-treated animals compared to vehicle (p<0.01), levetiracetam (p<0.01), and shams (p<0.05, *Figure 3—figure supplement 3B*). No significant differences were found in total tau protein expression.

## Sodium selenate-treated animals have longer telomere lengths

Telomere length was investigated as a surrogate marker of epilepsy severity and response to pharmacological treatment (*Chan et al., 2020*; *Galletly et al., 2017*; *Kang et al., 2020*; *Lukens et al., 2009*). Ear biopsies were taken 25 weeks post-SE to evaluate the effects of chronic epilepsy on telomere length. We found that TLE resulted in telomere shortening in vehicle-treated rats (p<0.001), and this was not prevented by levetiracetam treatment (p<0.01). However, sodium selenate treatment was able to reverse telomere shortening seen in post-SE animals treated with the vehicle (p<0.01), with the treated rats having telomere lengths similar to sham animals (*Figure 4*).

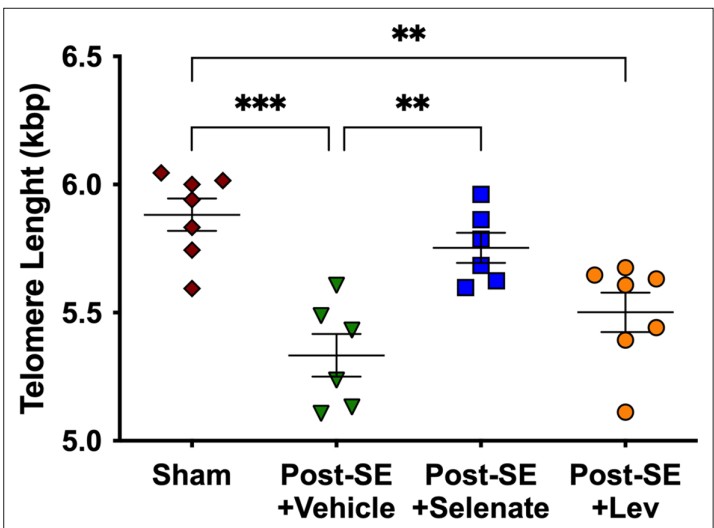

**Figure 4.** Sodium selenate prevents telomere shortening in drug-resistant temporal lobe epilepsy (TLE) rats. Chronic post-SE rats treated with levetiracetam and vehicle showed significantly reduced telomere length in comparison to sham. In contrast, sodium selenate-treated animals had significantly longer telomeres in comparison to the vehicle. kbp, kilobase pairs (*p<0.05, **p<0.01). One-way ANOVA with Tukey's post hoc. Data are shown as mean ± SEM.

## Seizure burden is correlated with shorter telomeres and poorer cognitive and sensorimotor outcomes

We also investigated the relationship between the seizure burden (i.e. number of seizures per day) at the end of the eight week post-treatment washout period, with the different outcomes of the behavioral tests and TL. We found that the number of seizures was inversely correlated with the preference for the novel object in the NOR ($r=-0.497$, $p<0.001$), NOP ($r=-0.357$, $p=0.016$), and TL ($r=-0.554$, $p=0.005$). The number of seizures was correlated with the search time in the MWM ($r=0.436$, $p=0.003$) and the number of slips and falls in the beam task ($r=0.762$, $p=0.001$).

## Network medicine integration highlights discrete protein-metabolite modules correlate with treatment and pre-clinical outcomes

Network medicine integration was performed to investigate pathways correlated with the TLE phenotype. We integrated the untargeted multi-omics (proteomics and metabolomics) datasets with the targeted proteomics and phosphoproteomics (PP2A, tau, and h-tau), TLE phenotype (seizures, cognitive, and sensorimotor outcomes), and telomere length studies to investigate protein-metabolite groups/pathways correlated with the response to selenate treatment. Weighted correlation network analysis identified 20 and 17 protein-metabolite modules, respectively, in the hippocampal and cortical multi-omic datasets. Several protein-metabolite modules showed significant positive and negative correlations with multiple treatment groups and various clinical measures (*Figure 5*). *Figure 5C* In the hippocampal network (*Figure 5A*), the Purple module shows significant inverse correlation with selenate treatment group, and is enriched for proteins and metabolites involved in aminoacyl-tRNA biosynthesis (FDR = 0.0059) and Lysine degradation (FDR = 0.021) pathways. This module is also inversely correlated with levetiracetam-treated group, but shows no correlation with the vehicle group. The Yellow module is significantly ($p=0.02$) correlated with the selenate-treated group and has no significant correlations with other groups. This module is enriched for proteins and metabolites involved in complement cascade, oxidative phosphorylation and other metabolic pathways.

In the somatosensory cortex correlation network (*Figure 5B*), the Tan ($p=0.002$), Magenta ($p=0.002$), Yellow ($p=0.02$), and Light-cyan ($p=0.02$) modules were significantly correlated with the selenate treated group and showed no correlation with the vehicle-treated group (*Figure 5B*). The magenta module does not annotate to any significant pathways; however, the Yellow module is enriched for proteins and metabolites involved in oxidative phosphorylation and other metabolic pathways, and the Tan module annotates to the insulin signaling pathway. Lastly, proteins and metabolites in the Light-cyan module are enriched for fatty acid biosynthesis pathways.

## Discussion

Current pharmacological treatments for TLE, ASMs, are merely symptomatic, attenuating seizures when taken on an ongoing basis without any sustained disease-modifying effects on the propensity to have seizures, or on its disabling associated neuropsychiatric, cognitive, and sensorimotor comorbidities (*Brooks-Kayal et al., 2013*; *Pitkänen et al., 2013*; *Saletti et al., 2019*). Once these treatments are discontinued (or the patient fails to take doses), the seizures relapse to at least the same severity and frequency as pre-treatment (*Coan and Cendes, 2013*; *Kwan et al., 2011*). Here, we report for the first time a pharmacological treatment that has persistent disease-modifying effects in an animal model of chronic drug-resistant TLE. Sodium selenate treatment was shown to have an enduring effect, mitigating the established epileptic state, and reducing the frequency of seizures and cognitive comorbidities. Our group has previously shown that treatment with sodium selenate in epilepsy models reduces h-tau, increases PP2A, and mitigates the development of epilepsy when delivered immediately after an epileptogenic brain injury before the occurrence of spontaneous recurrent seizures (i.e. an anti-epileptogenic effect) (*Jones et al., 2012*; *Liu et al., 2016*). However, the most common clinical scenario is that patients present to the clinic with established TLE, weeks to years after the epileptogenic brain insult, and in some patients, there is no identifiable brain insult (*Foster et al., 2019*).

The data from this study demonstrates that four weeks of sodium selenate treatment is associated with an enduring effect on mitigating the chronic epileptic state, reducing the frequency of seizures, neurocognitive, sensorimotor and neurobehavioral comorbidities, and biomarkers of disease severity.

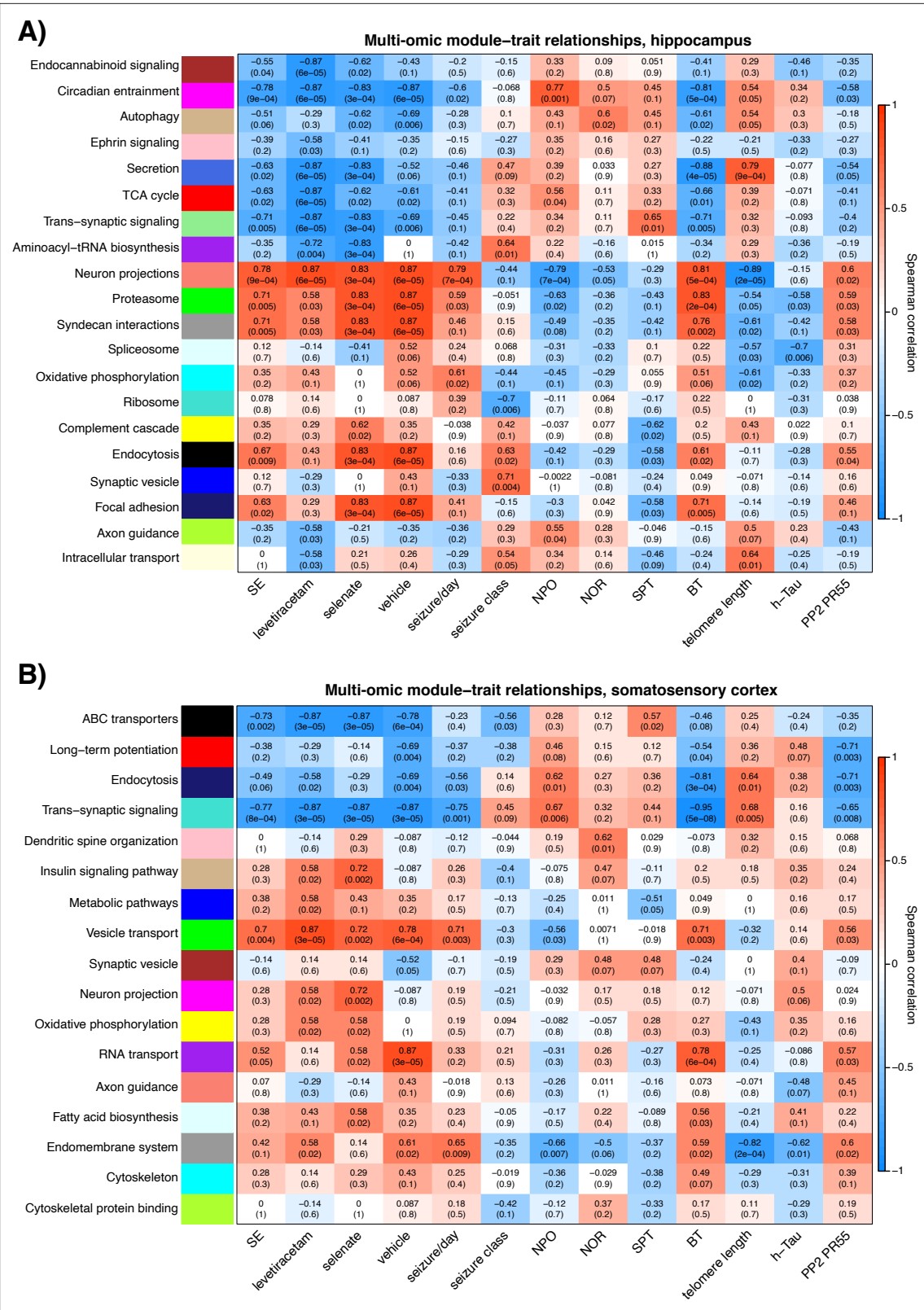

**Figure 5.** *Hippocampal and somatosensory cortex module-trait correlation heatmaps* . The relationship of protein-metabolite modules (in the Y-axis) with measured clinical traits (X-axis) in (**A**) the hippocampus and (**B**) somatosensory cortex. Modules are labeled by color and representative enriched pathway. Each block of the heatmap shows the Spearman correlation coefficient (SCC) of every module with clinical traits as well as the associated P value in brackets. The SCC values range from –1 to +1, depending on the strength and direction of the correlation.

Strikingly, 4 of 12 animals (33%), did not have any seizures at all at this final timepoint eight weeks following the cessation of the treatment. In contrast, all levetiracetam and vehicle-treated animals showed spontaneous seizures at the end of the study. It is important to note that, analogous to human chronic TLE, spontaneous remission of seizures is not part of the natural history of the post-KA SE rat model of chronic TLE (*Van Nieuwenhuyse et al., 2015a*; *Williams et al., 2009*).

Behavioral comorbidities are considered an integral part of the TLE disease (*Brooks-Kayal et al., 2013*; *Mazarati et al., 2018*). Patients with chronic and drug-resistant TLE have an increased prevalence of psychiatric conditions, memory, and learning disabilities (*Kandratavicius et al., 2012*; *Ott et al., 2003*; *Stafstrom et al., 1993*; *Stores, 1971*; *Tellez-Zenteno et al., 2007*). Cognitive impairment is the most common comorbidity that afflicts people with TLE and tends to resist treatment, even after epilepsy surgery (*Hermann et al., 2000*). In fact, in a proportion of cases, the cognitive dysfunction progresses despite successful seizure control (*Amlerova et al., 2014*; *Berg et al., 2012*; *Smith et al., 2014*; *Taylor et al., 2010*). In keeping with the human condition, cognitive deficits have been demonstrated as a comorbidity in the post-SE animal model (*Detour et al., 2005*; *Pearson et al., 2014*; *Chauvière et al., 2009*). Here, we found that animals treated with sodium selenate had improved spatial memory and recognition memory, comparable to non-epileptic controls. This is the first time that a treatment has been described to reverse memory deficits in established chronic TLE.

Sensorimotor deficits are another significant comorbidity in people with TLE (*Girardi-Schappo et al., 2021*; *Gandelman-Marton et al., 2006*), and are not necessarily associated with the presence of seizures or with the use of ASMs (*Gandelman-Marton et al., 2006*; *Petty et al., 2010*; *Shiek Ahmad et al., 2015*). A study in patients with drug-resistant TLE found abnormal postural instability and an increased risk of falls than healthy controls (*Gandelman-Marton et al., 2006*). Interestingly, transgenic mouse models that expressed h-tau have been described to not only present cognitive dysfunction but also sensorimotor impairments (*Kreilaus et al., 2021*; *Leroy et al., 2007*; *Lewis et al., 2000*). Here, we found that sodium selenate treatment mitigated the sensorimotor deficits in chronically epileptic rats in the post-KA SE model of TLE.

The results from this study add to the growing literature that indicates that pathological h-tau is a key pathophysiological driver of drug-resistant TLE and associated comorbidities. To further test this hypothesis, we performed untargeted proteomics using DIA-MS and targeted h-tau and PP2A protein analysis using Western Blots. Our data indicate that the persistent disease-modifying effects of sodium selenate in drug-resistant TLE correlate to the increased the expression of the B catalytic subunit of PP2A and the reduction of h-tau. Evidence has shown that the PP2A B subunit is specifically involved in the dephosphorylation of tau (*Barnes et al., 1995*; *Corcoran et al., 2010a*) and PP2A expression and activity are potentiated by sodium selenate (*Corcoran et al., 2010a*; *Jones et al., 2012*; *Shultz et al., 2015*).

In accordance with animal models (*Corcoran et al., 2010b*; *Jones et al., 2012*; *Shultz et al., 2015*), increased levels and deposits of h-tau have been described in surgically resected tissue from patients with drug-resistant TLE (*Puvenna et al., 2016*; *Sánchez et al., 2018*; *Sen et al., 2007*; *Tai et al., 2016*; *Thom et al., 2011*). In resected tissue from patients with drug-resistant TLE, PP2A protein expression is also significantly decreased, but remains unchanged in Alzheimer's disease patients, indicating that PP2A alterations may be an epilepsy-specific pathology (*Gourmaud et al., 2020*). There is also building evidence that tau-based mechanisms enhance neuronal excitability. Alterations in tau phosphorylation have been implicated in changing the excitation/inhibition balance in the neurons (*Palop et al., 2007*). Unstable microtubule assembly at axonal segments, as seen with the presence of h-tau, dysregulates the resting membrane potential and thereby generations of action potentials (*Matsumoto and Sakai, 1979*). Tau is transported into the cell via Fyn kinase where it influences N-methyl-D-aspartate (NMDA) receptor-mediated excitotoxicity (*van Eersel et al., 2010*) and changes in tau has been correlated with dysfunction of axonal NMDA receptors which are all considered mechanisms that promote the expression of seizures (*Liu et al., 2017*).

Pathological h-tau, has been proposed as one of the main culprits underlying cognitive deficits across different neurodegenerative disorders such as Alzheimer's disease, TBI, and epilepsy (*Casillas-Espinosa et al., 2020*). Accumulation of h-tau in temporal lobe brain tissue from patients with drug-resistant TLE is associated with cognitive decline (*Gourmaud et al., 2020*; *Thom et al., 2011*). Furthermore, tau pathology has been associated with a decline in verbal learning, recall, and graded

naming test scores in different neurodegenerative disorders (*Tai et al., 2016*) and in animal models that overexpressed h-tau (*Kreilaus et al., 2021*; *Leroy et al., 2007*; *Lewis et al., 2000*).

It is interesting that for some of the animals, the effects of sodium selenate were only observed several weeks after cessation of treatment, and the magnitude of the decrease in seizure expression progressively increased over the eight weeks of follow-up post-treatment. One possibility could be that the processes required to mitigate epileptic activity in neurons take time, particularly in chronically epileptic animals. However, it is also possible that the effects of sodium selenate extend beyond the described modification of h-tau and PP2A.

Dimension reduction is crucial for hypothesis generation to elucidate pathological mechanisms and changes in biological pathways as a response to treatment. To further explore sodium selenate's mechanism of action, we performed a network medicine integration of our untargeted/targeted multi-omics datasets, TLE outcomes (seizures and behavior), and molecular studies (western blots and telomere length analyses) to evaluate the correlation coefficients of single modules between the different treatment groups (e.g. Selenate and Vehicle). By mapping significant genes/proteins together with metabolites to biological pathways, our network integration approach allowed for the reduction in experimental and biological noise and higher confidence levels, thereby generating potential new disease modules (proteins and metabolites and the cellular pathways they are involved in). We found modules that constitute the 'healthy state,' which operate in a concerted fashion in a healthy brain but are lost in the setting of TLE. For example, the *Green Module* is positively correlated to epilepsy phenotype, poor cognitive performance, poor balance, shorter telomeres, decreased PP2A, increased h-tau expression, and cardiomyopathies, which are associated with several epilepsy syndromes (*Devinsky et al., 2018*; *Gilchrist, 1963*; *Naggar et al., 2014*; *Stöllberger et al., 2011*; *Surges and Sander, 2012*). Of particular interest are the *Yellow Module* in both the hippocampal and somatosensory cortex correlation networks as well as *Tan* and *Light-cyan Modules* in the somatosensory cortex. All of the above-mentioned modules positively correlated with selenate treatment. The *Yellow Module* is enriched for pathways involved in ATP metabolism, oxidative phosphorylation, and several inflammatory systems. The *Tan Module* is enriched for insulin signaling pathways and the *Light-cyan Module* annotates to fatty acid metabolism pathways. It is important to note that 'enrichment' does not mean 'increase' or 'upregulation.' The enrichment in insulin signaling, for example, highlights the involvement of this pathway in the epilepsy phenotype and selenate treatment, but unfortunately provides no information about the direction of change (upregulation or downregulation). In an attempt to capture changes in the regulation of biological pathways that may be influenced by sodium selenate treatment, we conducted a quantitative enrichment analysis on a pathway level (PADOG) between the selenate and vehicle groups, utilizing the Reactome pathway database resource. Based on the PADOG analysis results, there is significant upregulation of IGFR1 and FGFR signaling cascades and FGFR-mediated PI-3K cascade activation in the selenate-treated group compared to the vehicle. Given the neuroprotective role of PI-3K signaling in several epilepsy syndromes, it is conceivable that this is one of the potential mechanisms by which sodium selenate exerts its long-term disease-modifying effect. The strong correlation of the *Tan Module*, which annotates to insulin signaling pathways to selenate treatment only further highlights the involvement of insulin signaling in the potential mechanism of action of this treatment. The downregulation of fatty acid oxidation processes in the selenate group suggests a shift from hyperactive neurons with high-energy demand (like in epileptic seizures) towards neurons with lower energy demand.

To our knowledge, this analysis is the first attempt at the integration of metabolomic and proteomic data in an epilepsy network medicine paradigm. We demonstrate that multi-omic, data-driven correlation networks can be used for agnostic, hypothesis-free examination of disease states as well as the identification of specific protein-metabolite modules associated with different states and treatments. Further investigation of these pathways as potential mechanisms involved in the mode of action of sodium selenate can be instrumental in understanding and reversing the drug resistance of TLE.

Prognostic biomarkers are used to predict disease characteristics and identify disease recurrence or progression in patients who have the disease of interest. Unfortunately, no such prognostic biomarkers for TLE currently exist (*Simonato et al., 2021*). Telomere length has emerged as a potential prognostic biomarker of chronic neurodegenerative conditions (*Chan et al., 2020*; *Kang et al., 2020*; *Galletly et al., 2017*; *Lukens et al., 2009*). Telomeres are repetitive non-coding DNA-tandem sequences of TTAGGG found at the end of eukaryotic chromosomes (*Hehar and Mychasiuk, 2016*).

Telomere length has been used as a biological marker for cellular aging and has been described in many neurological disorders (*Hehar and Mychasiuk, 2016*; *Koppelstaetter et al., 2009*; *Sun et al., 2019*; *Wright et al., 2018*), including TLE (*Miranda et al., 2020*). Here, we analyzed telomere length from ear biopsies, as previous evidence has shown a strong correlation between telomere length in the brain and peripheral skin cells (*Hehar and Mychasiuk, 2016*). We found that telomere length was significantly reduced in TLE animals treated with vehicle or levetiracetam, and sodium selenate-treated animals had longer telomeres. The magnitude of the reduced telomere length was strongly associated with the number of seizures and more pronounced cognitive deficits in our drug-resistant TLE model. Overall, our data highlights the potential of telomere length as a prognostic biomarker of disease severity, treatment response, and TLE disease modification, which has the potential to be used in further trials of DMT.

Future studies should incorporate multi-omics and telomere length analysis integration performed at different timepoints from brain tissue and biofluids (blood and CSF) to map out the development, progression of epilepsy, and treatment response in this model. Unlike individual markers, pathway/network-based prognostic markers should be inherently more reliable since they provide the biological interpretation for the association between multiple molecules and the disease characteristics (seizures, behavior comorbidities, development of TLE) (*Zou et al., 2013*). Similarly, this approach could help to increase our understanding of the mechanism of action of sodium selenate as a disease-modifying treatment for established drug-resistant TLE.

## Conclusions

This study reports novel data that demonstrates that sodium selenate treatment for four weeks has disease-modifying effects in an animal model of chronic drug-resistant TLE. Even though, different pre-clinical treatments, including sodium selenate, have previously been described to have anti-epileptogenic effects immediately after the epileptogenic brain injury in animal models (*Casillas-Espinosa et al., 2019b*; *Liu et al., 2016*; *Huang et al., 2019*), this is the first time a pharmacological treatment has been shown to have disease-modifying properties in chronically epileptic animals. This approach is highly translatable, given that sodium selenate has already been shown to be safe and well tolerated in early-phase clinical trials for other conditions (*Cardoso et al., 2019*; *Corcoran et al., 2010b*; *Malpas et al., 2016*; *Vivash et al., 2020*), and there are a large number of patients with chronic drug-resistant TLE. The findings of this study, if confirmed in clinical trials, could be paradigm-shifting for the treatment of this common and highly disabling disease.

## Materials and methods
### Animals

Eleven-week-old male Wistar rats were used in all of the experiments. All procedures were approved by the Florey Animal Ethics Committee (ethics number 16–042 UM), adhered to the Australian code for the care and use of animals for scientific purposes, and were consistent with ARRIVE 2.0 guidelines (*Percie du Sert et al., 2020*). All the experiments were conducted by researchers blinded to the experimental conditions.

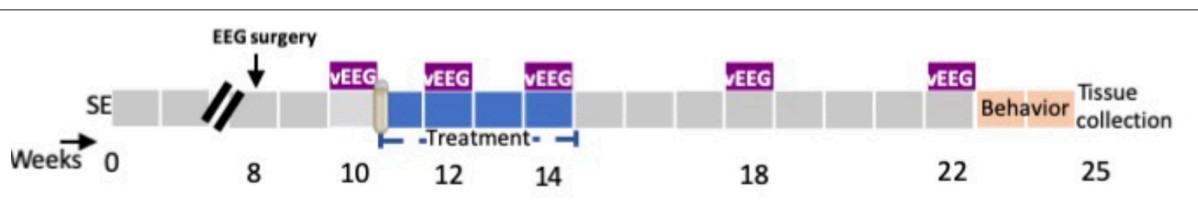

**Figure 6.** Experimental protocol in the post-status epilepticus model of temporal lobe epilepsy (TLE). SE = KA-induced status epilepticus, vEEG = video-electroencephalogram; weeks = weeks post-SE.

## Modified kainic acid-induced post-status epilepticus experimental protocol

The KA-induced post-SE model was used to evaluate the disease-modifying effects of sodium selenate. *Figure 6* summarizes the experimental paradigm for the current study.

The animals where individually housed with alternating 12 hr cycles of light and dark (lights on at 07:00 hr). Food and water were provided ad libitum for the whole duration of the study. A repeated low-dose KA administration protocol was used (*Bhandare et al., 2017*; *Casillas-Espinosa et al., 2019b*). Rats were injected i.p. with an initial dose of KA 7.5 mg/kg. Animals were monitored for behavioral seizures based on the Racine scale (*Racine, 1972*). If no continuous seizure activity was observed with at least five class five seizures of Racine, another i.p. dose of 5 mg/kg followed by 2.5 mg/kg doses of KA were administered up to a maximum of 20 mg/kg (*Brady et al., 2019*). An animal was eliminated from the experiment if it didn't show a stable self-sustained SE after a maximum KA dose. Shams were handled identically but received saline injections instead of KA. After 4 hr of sustained SE, **as evaluated by visual confirmation of behavioral seizures,** the animals were given diazepam (5 mg/kg/dose) to stop the SE (*Bhandare et al., 2017*; *Casillas-Espinosa et al., 2019b*).

## EEG electrode implantation surgery

Nine weeks after the induction of SE, rats were anesthetized with isoflurane (Ceva isoflurane, Piramal Enterprises Limited, India) and EEG Surgery was performed under an aseptic technique as previously described (*Casillas-Espinosa et al., 2019a*). (*Casillas-Espinosa et al., 2015*). Four burr holes were drilled through the skull without penetrating the dura, one on each side of the frontoparietal region (AP: ± 1.7; ML: –2.5), two to each side of the temporal region (AP: ± 5. 6; ML: left 2.5) anterior to lambda and four epidural stainless-steel screw recording electrodes (EM12/20/SPC, Plastics One Inc) were screwed into the burr holes. Ground and reference electrodes were implanted on each side of the parietal bone above the cerebellum (*Casillas-Espinosa et al., 2019c*; *Santana-Gomez et al., 2019*) and were secured using dental cement (VX-SC1000GVD5/VX-SC1000GMLLQ, Vertex, Australia) around the electrodes and over the skull. Buprenorphine (0.05 mg/kg, Indivior Australia) was used as an analgesic (*Casillas-Espinosa et al., 2019c*).

## Video-EEG acquisition

Ten weeks after the induction of SE, animals were connected using cables (M12C-363, Plastics One Inc, Australia) (*Casillas-Espinosa et al., 2019b*). vEEG recordings were acquired using Profusion 3 software (Compumedics, Australia) unfiltered and digitized at 512 Hz.

## Pre-treatment seizure analyses screening

All vEEG recordings were screened for seizures using Assyst, a semi-automated seizure detection software (Assyst, Australia) (*Casillas-Espinosa et al., 2019a*). Seizure events were visually confirmed using Profusion 3 software by two blinded independent experts. A seizure was defined as an episode of rhythmic spiking activity that was three times the baseline amplitude and a frequency >3 Hz that lasted at least 10 s; the end of a seizure was determined as the last spike (*Brady et al., 2019*; *Casillas-Espinosa et al., 2019a*; *Casillas-Espinosa et al., 2019b*; *Ndode-Ekane et al., 2019*; *Pitkänen et al., 2005*; *Van Nieuwenhuyse et al., 2015b*). The average number of seizures per day, average seizure duration, and seizure class (i.e. severity, Racine scale) were analyzed (*Brady et al., 2019*; *Casillas-Espinosa et al., 2019b*). Animals that did not have two or more seizures during the week of baseline video-EEG (vEEG) were discarded from the experiments.

## Group allocation and osmotic pump implantation surgery

Shams (n=11); and post-SE rats that had seizures during the pre-treatment seizure screening week were randomly allocated to one of the four treatment groups: post-SE +vehicle (n=12), post-SE +sodium selenate (n=12), post-SE +levetiracetam (n=12). Subcutaneous osmotic pumps (2ML4, delivery rate 2.5 µl/h, Alzet, Cupertino, CA, USA) that delivered sodium selenate (1 mg/kg/day; SKU: 71948, Sigma-Aldrich, Castle Hill, Australia), levetiracetam (200 mg/kg/day; Angene, London, England) or vehicle (saline 0.9%) were implanted and treatment was continuously delivered for four weeks (*Casillas-Espinosa et al., 2019b*; *Li et al., 2018*). Sodium selenate and levetiracetam doses

were selected based on previous studies (*Brandt et al., 2007*; *Casillas-Espinosa et al., 2019b*; *Li et al., 2018*; *Löscher and Hönack, 1993*; *Löscher et al., 1998*; *Shultz et al., 2015*).

## vEEG analysis to evaluate disease modification in chronic TLE

To evaluate the disease-modifying effects of the different treatments, animals underwent seven days of continuous vEEG recordings during the second and fourth week of treatment, and during the weeks four and eight post-washout (*Figure 1*). A washout period of eight weeks was chosen to provide a long-term monitoring of the disease-modifying effects of the treatments (*Barker-Haliski et al., 2015*; *Casillas-Espinosa et al., 2019b*). Seizure duration and class were considered from only the animals that presented with spontaneous seizures. vEEG analysis was performed in a blinded manner and confirmed by two different observers as described above.

## Behavioral tests

All behavioral tests were performed in a light-controlled (~110 lux), closed, quiet, and clean room between 9 am and 5 pm. The animals had at least 1 hr to acclimatize to the room before the start of any of the behavioral tests. Behavioral testing was performed in a blinded manner to the treatment groups. All of the behavioral tests were video recorded for future analysis. Open field (OF), elevated plus maze (EPM), Novel object placement (NOP), Novel Object Recognition (NOR), beam test (BT), and Morris water maze (MWM) sessions were filmed using a digital camera mounted above the center of the arena, which was connected to a computer for the recording and objective analysis of digitized behavioral tests (Ethovision 3.0.15, Noldus) and was conducted as previously described (*Bao et al., 2012*; *Broadbent et al., 2004*; *Casillas-Espinosa et al., 2019b*; *Johnstone et al., 2015*; *Jones et al., 2008*; *Kolb and Whishaw, 1985*; *Morris, 1984*; *Pearson et al., 2014*; *Shultz et al., 2015*).

### Elevated plus maze (EPM)

The EPM is a custom-made black acrylic arena in the shape of a plus, with one opposing pair of arms enclosed by 30 cm high walls (closed arms) and the other opposing arms without any enclosure (open arms). Each arm is 13 cm wide and 43.5 cm long, and the maze is elevated 60 cm above the floor. The rat was placed in the center of the EPM, and its behavior was monitored over 5 min (*Casillas-Espinosa et al., 2019b*; *Johnstone et al., 2015*; *Jones et al., 2008*). The EPM was wiped clean between each animal. The distance traveled, the number of entries into the open and closed arms, and the total time spent in each arm during the trial were recorded.

### Open field (OF)

The OF is a 100 cm diameter circular arena, with an inner circle arena of 66 cm of diameter that was virtually defined using the Ethovision software. For each test, the rat is placed gently into the center of the field and its behavioral activity is monitored for 10 min. The distance traveled as well as the entries and time spent in the inner circle were recorded (*Casillas-Espinosa et al., 2019b*; *Johnstone et al., 2015*; *Jones et al., 2008*).

### Novel object placement (NOP)

The test took place in the same circular arena used for the OF test, and consisted of two phases, a learning phase, and a memory (testing) phase. During the learning phase, animals were placed into the behavioral arena for a period of 5 min and allowed to explore two identical stimulus objects (plastic green trees) before being placed back into the home cage. 15 min after the first stage, the animal was re-introduced to the arena, except that during the memory phase, one of the objects was displaced to a novel spatial location. In the first stage, two identical objects (plastic green tree) were each placed at the center of two adjacent quadrants of the circular arena and rats were allowed to freely explore the arena for another 5 min (*Broadbent et al., 2004*; *Pearson et al., 2014*). Animals were assessed on the time spent around the objects, and the percentage of time the animal investigated the object in a novel location.

### Novel object recognition (NOR)

The first phase of the test was identical to the learning phase of the NOP test. However, in the second stage of the NOR test, commencing 15 min after the first stage, one of the objects was replaced by a novel and unfamiliar object (a plastic red car), which was placed in the exact same position as one of the original objects and the animal was allowed to explore the arena for 5 min (*Broadbent et al., 2004*; *Pearson et al., 2014*). Animals were assessed on the time spent around the objects, and the percentage of time the animal investigated the novel object.

### Beam task (BT)

The beam task (BT) is used to assess sensorimotor function (*Bao et al., 2012*; *Kolb and Whishaw, 1985*). The test was divided into a training and a testing session. During the training session, rats underwent five trials to traverse a 100 cm long beam with a width of 4 cm, and a further five trials to traverse a 100 cm long beam with a width of 2 cm. Twenty-four hours after the training session, the BT testing session was performed and consisted of 10 trials on the 2 cm wide, 100 cm long beam. Each trial began with the rat being placed at one end of the beam and ended when the animal success-fully traversed a distance of 100 cm. A maximum of 60 s was allowed for each trial. If the rat fell off the beam it was given a time of 60 s. Each trial was recorded by a camera, and traverse time and the number of slips and falls were used as measures of sensorimotor function (*Shultz et al., 2015*).

### Morris water maze (MWM)

The MWM is a circular black PVC plastic pool of 160 cm diameter which is filled to a depth of 2.5 cm above a 10 cm diameter is hidden platform water at 25 ± 1°C (*Casillas-Espinosa et al., 2019b*; *Morris, 1984*; *Shultz et al., 2015*). The platform is located in the center of the South-West quadrant. Visual cues are placed in the main four cardinal points, consisting of large black and white symbols that serve as orientation landmarks for the rats. The test was performed over two days in an acquisi-tion (day 1) and reversal (day 2) session. The acquisition session consisted of 10 trials. A **trial** begins by placing gently the rat in the pool, facing the pool wall, and ended when the rat stands on the platform for at least 3 s, or when 90 s had elapsed. Each trial began at one of four pool wall start locations (North, South, East, or West) according to a pseudo-random schedule of start locations that prevented repeated sequential starts from the same location (*Liu et al., 2016*; *Shultz et al., 2015*). The reversal session of the water maze was completed 24 hr after acquisition. The procedures were identical to those of the acquisition session except that the hidden platform was now located in the north-west quadrant of the pool. The latency to find the escape platform, speed, and strategic path used by the animal to locate the platform are recorded. Search time was used as measure of spatial orientation and memory (*Bao et al., 2012*; *Casillas-Espinosa et al., 2019b*; *Morris, 1989*; *Whishaw and Jarrard, 1995*). Swim speed (cm/s) was used as a measure of motor ability (*Bao et al., 2012*).

### Sucrose preference test (SPT)

The SPT is a well-validated measure of depression/anhedonia in rats (*Casillas-Espinosa et al., 2019b*; *Jones et al., 2008*). The SPT was used to evaluate the effects of the treatments to modify the depressive-like behavior seen in post-SE rats. The sucrose preference was calculated as the percentage of the sucrose solution consumed from the total volume (of both sucrose and water). The animals remained in their same home cage during the testing. One hour before the start of the test, the animals were given up to 0.5 ml of the testing 2% sucrose to ensure that the animals would drink the sucrose solution. The animals were presented with two bottles, one filled with water and the other one filled with 2% sucrose in water for 24 hr (*Casillas-Espinosa et al., 2019b*). Bottle position was randomized to avoid position preference (*Casillas-Espinosa et al., 2019b*; *Jones et al., 2008*; *Sarkisova et al., 2003*). Total fluid intake and percentage preference for sucrose were recorded. The results of the sucrose preference test are presented in *Figure 2—figure supplement 2*.

### Tissue collection

Seven days after the last behavioral test, the animals were anesthetized using 5% isoflurane, and were culled using a lethal dose of intraperitoneal Lethabarb (150 mg/kg IP; pentobarbitone sodium; Virbac, Australia) (*Casillas-Espinosa et al., 2019b*). Tissue was taken from each rat's ear and stored at

−80 °C for telomere length (TL) analysis (*Hehar and Mychasiuk, 2016*). Using the Rat Brain Atlas as a reference (*Paxinos and Watson, 1982*), the hippocampus and somatosensory cortex from (n=3) rats per treatment cohort were rapidly dissected, flash frozen with liquid nitrogen, and stored at −80 °C for untargeted proteomics, metabolomics, and western blot studies.

## Data-independent proteomic analysis

Frozen hippocampus and somatosensory cortex samples were homogenized in liquid nitrogen and lysed in 4% SDS, 100 mM HEPES, pH8.5, which was then heated at 95 °C for 5 min and sonicated three times for 10 s each at an amplitude of 10 µm. The lysate was clarified by centrifuging at 16,000 g for 10 min and protein concentration was determined by Pierce BCA Protein Assay Kit (Thermo). Equal amount of protein was denatured and alkylated using Tris(2-carboxyethyl)-phosphine-hydrochloride (TCEP) and 2-Chloroacetamide (CAA) at a final concentration of 10 mM and 40 mM, respectively, and incubated at 95 °C for 5 min. SDS was removed by chloroform/methanol. Sequencing grade trypsin was added at an enzyme-to-protein ratio of 1:100 and the reaction was incubated overnight at 37 °C. The digestion reaction was stopped by adding formic acid to a concentration of 1%. The peptides were cleaned with BondElut Omix Tips (Agilent) and concentrated in a vacuum concentrator prior to analysis by mass spectrometry.

Using a Dionex UltiMate 3000 RSLCnano system equipped with a Dionex UltiMate 3000 RS autosampler, the samples were loaded via an Acclaim PepMap 100 trap column (100 µm × 2 cm, nanoViper, C18, 5 µm, 100 å; Thermo Scientific) onto an Acclaim PepMap RSLC analytical column (75 µm × 50 cm, nanoViper, C18, 2 µm, 100 å; Thermo Scientific). The peptides were separated by increasing concentrations of 80% ACN/0.1% FA at a flow of 250 nl/min for 158 min and analyzed with a QExactive HF mass spectrometer (Thermo Scientific) operated in data-independent acquisition (DIA) mode. Sixty sequential DIA windows with an isolation width of 10 m/z have been acquired between 375–975 m/z (resolution: 15.000; AGC target: 2e5; maximum IT: 9ms; HCD Collision energy: 27%) following a full ms1 scan (resolution: 60.000; AGC target: 3e6; maximum IT: 54ms; scan range: 375–1575 m/z). The acquired DIA data have been evaluated in Spectronaut 13 Laika (Biognosys) using an existing, in-house generated spectral library derived from rat brain samples.

## Untargeted metabolomics studies

The same frozen hippocampal and somatosensory cortex samples used for proteomics were cryogenically pulverized (cryopulverization) using a 12-well Biopulverizer (BioSpec Products, OK USA Part number 59012 MS) according to the manufacturer's instructions. The frozen tissue powder was then weighed and extracted in 20 µL of extraction solvent (0 °C) per mg of tissue. The mixture was then briefly vortexed before sonication in an ice-water bath for 10 min followed by centrifugation (20,000 rcf, 4°C, 10 min). The supernatant was then transferred to a mass spectrometry vial for LC-MS analysis. The extraction solvent consisted of 2:6:1 CHCl3:MeOH:H2O v/v/v with 2 µM CHAPS, CAPS, PIPES, and TRIS as internal standards.

Plasma (25 µL) was extracted by addition of 200 µL of 1:1 acetonitrile:MeOH v/v with 2 µM CHAPS, CAPS, PIPES, and TRIS as internal standards at 0 °C. Samples were then mixed on a vortex mixer for 45 cycles 20 s each at 4 °C after which they were centrifuged (20,000 rcf, 4°C, 15 min) and the supernatant was then transferred to a mass spectrometry vial for LC-MS analysis.

For LC−MS analysis, samples were analyzed by hydrophilic interaction liquid chromatography coupled to high-resolution mass spectrometry (LC−MS) (*Creek et al., 2012*; *Srivastava et al., 2017*). In brief, the chromatography utilized a ZIC-pHILIC column (column temperature 25 °C) with a gradient elution of 20 mM ammonium carbonate (A) and acetonitrile (B) (linear gradient time-%B as follows: 0 min-80%, 15 min-50%, 18 min-5%, 21 min-5%, 24 min-80%, 32 min-80%) on a Dionex RSLC3000 UHPLC (Thermo). The flow rate was maintained at 300 µL/min. Samples were kept at 4 °C in the autosampler and 10 µL injected for analysis. The mass spectrometry was performed at 35,000 resolution (accuracy calibrated to <1 ppm) on a Q-Exactive Orbitrap MS (Thermo) operating in rapid switching positive (4 kV) and negative (−3.5 kV) mode electrospray ionization (capillary temperature 300 °C; sheath gas 50; Aux gas 20; sweep gas 2; probe temp 120 °C). All samples were analyzed in randomized order and with pooled quality control samples analyzed regularly throughout the batch to confirm reproducibility. Approximately 300 metabolite standards, including all reported amino acids,

were analyzed immediately preceding the batch to determine accurate retention times to confirm metabolite identification.

For data analysis, untargeted metabolomics data were analyzed using the IDEOM (version 20) workflow with default parameters (*Creek et al., 2012*). In brief, this involved peak picking with XCMS (*Tautenhahn et al., 2008*), peak alignment and filtering with mzMatch (*Scheltema et al., 2011*), and further filtering, metabolite identification, and comparative analysis with IDEOM.

## Proteomic and metabolomic data analysis

The proteomic and metabolomic mass-spectroscopy data from the somatosensory cortex and hippocampus of rats treated with selenate, levetiracetam, vehicle, and sham were tabulated into spectral peak intensity values followed by log2 transformation and unit-variance scaled as we have described previously (*Harutyunyan et al., 2022*). The missing values were omitted by replacing them with a background signal. One-way ANOVA and Tukey's HSD were employed to identify differences in metabolite and protein abundance between levetiracetam, selenate, vehicle, and sham groups. A false discovery rate of FDR <0.05 was deemed significant. In addition to the univariate analysis, multivariate analysis in the form of principal component analysis (PCA) and partial least square-discriminant analysis (PLS-DA) was performed in order to identify the metabolites and proteins driving the separation between treatment groups (*Harutyunyan et al., 2022*). Metabolite set enrichment analysis (MSEA) was performed between all pairs of groups via the quantitative enrichment module of MetaboAnalyst 5.0 web tool (*Pang et al., 2021*) for the metabolomic data.

Quantitative gene set enrichment analysis was performed between all pairs of treatment groups via Pathway Analysis with Down-weighting of Overlapping Genes (PADOG) module of Reactome web-based tool (*Fabregat et al., 2018*; *Jassal et al., 2020*) for proteomic data. PADOG is a weighted gene/protein set analysis method that down-weighs genes/proteins that are present in many pathways thus preventing their exaggerated significance. Pathways with FDR of <0.05 were considered differentially regulated.

## Multi-omic correlation network analysis, module detection, and functional annotation

We employed early integration and weighted correlation network analysis (WGCNA) (*Zhang et al., 2013*; *Harutyunyan et al., 2022*) methods to characterize the proteome and metabolome of SE rats. Correlation-based networks of proteins and metabolites were constructed, which were then clustered into modules and functionally annotated through pathway enrichment analysis (*Harutyunyan et al., 2022*).

To integrate the proteomic and metabolomic data for network analysis, we unit variance scaled and concatenated the normalized proteomic and metabolomic data into a single matrix. Unit variance scaling uses standard deviation as the scaling factor; thus, the resultant integrated data can be analyzed on the basis of correlations (*Zhang et al., 2013*). Correlation-based multi-omics networks were then constructed for all treatment groups by employing the framework of WGCNA using the WGCNA R package (*Langfelder and Horvath, 2008*). Briefly, an adjacency matrix was constructed reflecting the pairwise Pearson correlations between all detected proteins and metabolites across all samples. The correlation network was then built based on the adjacency matrix, where each node corresponds to a single metabolite or protein, and the edges between the nodes represent the correlation between the relative abundance of the given metabolite/protein across all samples. A clustering algorithm was employed to identify metabolite/protein modules (labeled by color), which then were functionally annotated through joint pathway enrichment analysis (*Pang et al., 2021*) of all metabolites/proteins constituting each module. In addition to unique color labels, each module was labeled by a representative biological pathway based on functional enrichment analysis. The Spearman correlation of modules to clinical traits such as average seizure class, telomere length, and cognitive performance was examined.

## Targeted protein expression using automated capillary western blots

Capillary Western blots were used to assess protein expression of hyperphosphorylated tau (h-tau), total tau, and PP2A. A portion of the frozen somatosensory cortex\tissue was homogenized in radio-immunoprecipitation assay (RIPA) buffer with protease and phosphatase inhibitors (SigmaFast, S8820,

Sigma-Aldrich, USA and PhosStop, 4906837001, Sigma-Aldrich, USA) The protein concentration of the lysates was quantified using a BCA Protein Assay Kit (Thermo Scientific Pierce Biotechnology, USA) and Benchmark Plus Microplate Spectrophotometer (Bio-Rad, Hercules, United States). Capillary Western analyses were performed using the Wes Protein Simple Western System (12 kDA Wes separation module kit, SWM-W001, ProteinSimple, United States) according to the manufacturer's instructions. Individual assays were performed for total tau (Tau-5, # 577801, Millipore, Australia, 1:50), h-tau (p-Ser202/Thr205-tau (AT8), # MN1020, ThermoFisher Scientific, Australia, 1:200), PP2A subunit B (#05–592, Millipore, Australia, 1:200) and Glyceraldehyde 3-phosphate dehydrogenase (GAPDH; 1:200, Santa Cruz Biotechnology #sc-47724, USA) was used as a loading control. H-tau was determined as the ratio of phosphorylated tau (AT8) relative to total tau (tau 5) (*Shultz et al., 2015*; *Liu et al., 2016*). Raw or data is expressed as the percentage of the average of controls (*Liu et al., 2016*). All the semi-automated western blot procedures were conducted by an experimenter blinded to experimental conditions.

## Telomere length analysis

Ear samples (5–7/group were collected at the end of the study). The Qiagen DNA Micro kit (Qiagen, Germany) was used to extract genomic DNA (gDNA) as described in *Cawthon, 2002*; *Hehar and Mychasiuk, 2016*. gDNA samples had mean 260/230 and 260/280 spectral ratios of 2.24 and 1.90, respectively. TL analysis was conducted on all diluted DNA samples (20 ng/µL) using a similar protocol to that previously described (*Sun et al., 2019*). RT-qPCR reactions were conducted by adding 1 µL DNA, 1 × SYBR Green FastMix with Rox, and primers so that the total volume in each well was 20 µL. Each reaction was performed in duplicate, including no template controls (NTC) to ensure that the reactions were not contaminated. Primer final concentrations were (forward/reverse): 270 nM/900 nM for Tel; and 300 nM/500 nM for 36B4 (*Hehar and Mychasiuk, 2016*). Absolute quantitative PCR was used to determine the ratio of telomeres to a single copy gene (36B4), calculated as $[2Ct(telomeres) / 2Ct(36B4)]-1 = -2-\Delta Ct$. This was then used in the following formula to determine TL: $TL = 1910.5 (-2-\Delta Ct) + 4,157$ (*Cawthon, 2002*; *Hehar and Mychasiuk, 2016*; *Sun et al., 2019*).

## Statistical analysis

All EEG parameters and MWM were normally distributed as per the Shapiro-Wilk test. EEG data were compared using a two-way repeated measure ANOVA, separating the periods during and following the treatment period into different analyses (*Dezsi et al., 2013*). During treatment, the between subject factor was treatment, and the within subject factor was the timepoint of recording. After treatment (i.e. drug washout), ANOVA was used to assess whether seizures were modified following the removal of treatment, using the same parameters as above. *Post hoc* analyses were used to test for multiple comparisons and control for a chance. For the seizure data, Tukey's *post hoc* was used. MWM was also analyzed, using two-repeated measures ANOVA with Dunnett's *post hoc*. Rest of the data were analyzed for normality using the Shapiro-Wilk test. Variance was assessed using the Brown-Forsythe test. For non-normally distributed data or where variance was unequal, a $\log_{10}$ transformation was used (i.e. SPT, NOP) to resolve the heterogeneity. Normally distributed, (OF, EPM, NOR, BT, TL, and PP2A western blot) data were analyzed using one-way ANOVA with Tukey *post hoc*. For the data that did not fit a normal or Gaussian distribution (NOP, NOR, SPT, h-tau/total tau western blot), and was unequal after a $\log_{10}$ transformation (SPT) the non-parametric Kruskal-Wallis test with Dunn's *post hoc* was used instead. Correlation studies were performed on data that was significantly different between the experimental groups. Pearson correlation was performed between the average number of seizures per day, MWM, BT, and TL. Spearman correlation was performed between the average number of seizures per day, NPO and NOR. The analyzed data was reported as the mean and standard error of the mean (SEM) with statistical significance set at $p<0.05$. All the statistical comparisons were done using GraphPad Prism 9 (GraphPad Software, Inc USA).

## Sample size

Power analyses based on our previously published KA-induced post-SE rat model studies (*Casillas-Espinosa et al., 2019b*; *Liu et al., 2016*), indicated that an n=8 per group was required to detect a 30% difference between the treatment groups considering $\alpha=0.05$ and power = 95%. For the multi-omics experiments, a minimum of n=3 animals per group was used, based on a previous study in

which a combination of combining omics data from two experimental modalities, showed to be sensitive to identify meaningful biological insights (*Schouten et al., 2016*).

## Acknowledgements

This study used BPA-enabled (Bioplatforms Australia) / NCRIS-enabled (National Collaborative Research Infrastructure Strategy) infrastructure located at the Monash Proteomics & Metabolomics Facility led by RBS.

PMCE was funded by a NHMRC Early Career Fellowship (#APP1087172) and the University of Melbourne Early Career Research Grant (#603834). RM – Canadian Institutes for Health Research (CIHR) PTJ – 153051. SRS is supported by a NHMRC Level 2 Career Development Fellowship.

## Additional information

### Funding

| Funder | Grant reference number | Author |
| --- | --- | --- |
| National Health and Medical Research Council | APP1087172 | Pablo M Casillas-Espinosa |
| University of Melbourne | 603834 | Pablo M Casillas-Espinosa |
| CIHR Skin Research Training Centre | PTJ - 153051 | Richelle Mychasiuk |
| National Health and Medical Research Council | Level 2 CDF | Sandy R Shultz |

The funders had no role in study design, data collection and interpretation, or the decision to submit the work for publication.

### Author contributions

Pablo M Casillas-Espinosa, Conceptualization, Data curation, Formal analysis, Funding acquisition, Investigation, Methodology, Writing – original draft, Writing – review and editing; Alison Anderson, Data curation, Formal analysis, Writing – review and editing; Anna Harutyunyan, Ralf B Schittenhelm, Formal analysis, Methodology, Writing – review and editing; Crystal Li, Cheng Huang, Data curation, Investigation; Jiyoon Lee, Emma L Braine, Christopher K Barlow, Data curation; Rhys D Brady, Data curation, Writing – review and editing; Mujun Sun, Investigation; Anup D Shah, Data curation, Methodology; Richelle Mychasiuk, Data curation, Investigation, Methodology, Writing – review and editing; Nigel C Jones, Methodology, Writing – review and editing; Sandy R Shultz, Data curation, Methodology, Writing – review and editing; Terence J O'Brien, Conceptualization, Supervision, Visualization, Writing – review and editing

### Author ORCIDs

Pablo M Casillas-Espinosa ⓘ http://orcid.org/0000-0002-6199-9415
Anna Harutyunyan ⓘ http://orcid.org/0000-0002-6165-5639
Sandy R Shultz ⓘ http://orcid.org/0000-0002-2525-8775
Terence J O'Brien ⓘ http://orcid.org/0000-0002-7198-8621

### Ethics

All procedures were approved by the Florey Animal Ethics Committee (ethics number 16-042-UM), adhered to the Australian code for the care and use of animals for scientific purposes and consistent with ARRIVE 2.0 guidelines.

### Decision letter and Author response

Decision letter https://doi.org/10.7554/eLife.78877.sa1
Author response https://doi.org/10.7554/eLife.78877.sa2

# Additional files

## Supplementary files
- Supplementary file 1. Untargeted proteomic analysis from the somatosensory cortex.
- Supplementary file 2. Untargeted proteomic analysis from the hippocampus.
- Supplementary file 3. Untargeted metabolomic analysis from the somatosensory cortex.
- Supplementary file 4. Untargeted metabolomic analysis from the hippocampus.
- Transparent reporting form

## Data availability
All the metrodology and results form this work have been included in the manuscrit or as part of the supplementary material. Data has been deposited in Dryad https://doi.org/10.5061/dryad.37pvmcvnd.

The following dataset was generated:

| Author(s) | Year | Dataset title | Dataset URL | Database and Identifier |
|---|---|---|---|---|
| Casillas-Espinosa PM | 2022 | EEG and multi-omics data | https://dx.doi.org/10.5061/dryad.37pvmcvnd | Dryad Digital Repository, 10.5061/dryad.37pvmcvnd |

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
