## [Editor Report]

This important study provided evidence that sodium selenate is a treatment that reduces the symptoms of chronic seizures in a rat model of epilepsy. The improved symptoms were reduced seizure frequency, improved memory, and improved sensorimotor function. The authors provided convincing evidence that even after treatment ended the benefits persisted, a remarkable finding. They also identified several possible mediators of the effects, including PPAR and h-tau, as well as telomere length. An in-depth study of potential genetic regulation by selenate provided additional possible targets of selenate that ameliorate epilepsy.

---

## [Decision Letter]

**Decision letter after peer review:**

Thank you for submitting your article "Disease-modifying effects of sodium selenate in a rat model of drug-resistant, temporal lobe epilepsy" for consideration by *eLife*. Your article has been reviewed by 2 peer reviewers, one of whom is a member of our Board of Reviewing Editors, and the evaluation has been overseen by Christian Büchel as the Senior Editor. The reviewers have opted to remain anonymous.

Summary of findings and potential significance:

Sodium selenate reduced seizures when administered after initiation of epilepsy, complementing earlier work showing efficacy if administered before initiation. Data were compared to Levitiracetam and vehicle-treated animals. Sodium selenate further improved cognitive impairments. It also persistently reduced phospho-tau and increased PP2A protein expression, and reversed TLE-associated telomere-shortening. Finally, proteome and metabolome data from the animal model of epilepsy is discussed and provide initial insights into the effects of sodium selenate treatment on molecular pathology.

Essential revisions:

1) Address novelty/significance by providing evidence PP2A, tau, and telomeres are critical to the hypothesized mechanisms.

2) Clarify and extend molecular data as described by Reviewer 2.

3) Address lesser concerns itemized below.

*Reviewer #1 (Recommendations for the authors):*

To improve the paper, the authors should consider how additional experiments could make it novel and a significant advance. Defining the mechanism would be an example. Additional endpoints typically studied or additional models might be useful if they shed light on the mechanism. Clinical relevance would be increased by human data.

*Reviewer #2 (Recommendations for the authors):*

The multi-omics analysis appears to be the weakest part of the manuscript. Given the interindividual variability and complexity of proteomic and metabolomic data, an n of 3 per group is small. In the Materials and methods section, the authors mention a power analysis that suggested a group size of 8. Please comment.

The authors state that network medicine integration highlights discrete protein-metabolite modules that correlate with treatment and pre-clinical outcomes. This part needs much more attention and clarification. There is no individual discussion of the proteomic and metabolomic data including some supporting Figures (e.g., multivariate analysis/PCA, volcano plot, hierarchical clustering). Results from this section should not be "hidden" in the supplements.

Figure 5 is not intuitive. Please consider using different labels with a clear association to their gene/protein context, e.g, metabolism instead of the red module. Please provide more specific information on module size, gene content, and associations. A network graph (node size = module size, color code of nodes = Spearman correlation coefficient) may be used to help visualization. Are there any proteins or modules that distinguish treatment groups?

Figures 1-4: Please stick to a uniform color code across all figures, so that, e.g., Lev or Sodium selenate treated animals always have the same color in all graphs. The authors further may consider changing the labeling to support a more intuitive understanding of the figures, e.g., sham-control, post-SE+vehicle, post-SE+Lev, post-SE+Sodium selenate

[Editors' note: further revisions were suggested prior to acceptance, as described below.]

Thank you for resubmitting your work entitled "Disease-modifying effects of sodium selenate in a rat model of drug-resistant, temporal lobe epilepsy" for further consideration by *eLife*. Your revised article has been evaluated by Christian Büchel (Senior Editor) and a Reviewing Editor.

The manuscript has been improved but there are some remaining issues that need to be addressed, as outlined below:

The strength of the manuscript continues to be the persistent nature of sodium selenate to reduce seizures, in the KA post-SE model, even after treatment has stopped. The limitations of the paper continue to be the lack of evidence that the changes in molecules like PPAR and h-tau, or changes in telomeres, mediate the effects of selenate. The authors were asked to provide such evidence or make their conclusions more cautious and they do not appear to have done so. However, the data about PPAR and other molecules are interesting. The authors just need to refrain from conclusions about cause-effect. For example, the conclusions state that h-tau mediates the effects of selenate and that was not shown. Other statements along these lines also require revision.

The authors have improved the section about transcriptomics, but here there are mainly correlative data also, so while improved, there is a limit to the impact of the work.

Methodological concerns are:

Only 1 week of seizure analysis by video-EEG. The standard is 2 weeks and many investigators use a longer time because seizures in the post-SE models are not consistent from week-to-week. One reason is seizures cluster so that one week there can be very few seizures and the next week the opposite. This may have been the reason for the variability in the graphs. On the other hand, selenate treatment led to a marked reduction and lower variance by the end of the experimental period, making it unlikely that selenate lacked effects. One issue that should be addressed is how the authors handled the inequality of variance in their statistics. Usually it requires data transformation or non-parametric statistics.

Recommendations:

1. Remove all statements that PPAR, tau, telomeres, (etc.) and transcriptomics data caused effects of selenate. These data are only correlative.

2. Revise statistics so that where variance is unequal in the different groups all data are either transformed (common methods are to take the log or reciprocal of all data points) to resolve the heterogeneity (making parametric statistics possible to use), or (when transformation does not resolve heterogeneity of variance) non-parametric statistics are used. To test whether variance is unequal, Bartlett's test and Brown-Forsythe are used.

---

## [Author Response]

Essential revisions:1) Address novelty/significance by providing evidence PP2A, tau, and telomeres are critical to the hypothesized mechanisms.

We thank the reviewers and editors for their comments. While previous work from our group has described that sodium selenate activates the PP2A subunit B (Corcoran et al., 2010) and reduces phosphorylated tau after traumatic brain injury and administration of KA but before the occurrence of spontaneous seizures (*i.e.* before epilepsy has been established) (Liu et al., 2016, Shultz et al., 2015). Here, we show that sodium selenate is disease-modifying if given in the chronic established epilepsy state (i.e. 10 weeks post-KA administration). This is of great potential clinical relevance, as most commonly patients present to the neurologist in practice after the epilepsy has been established, not in the pre-epileptic state.

To further investigate the mechanism by which sodium selenate exerts its disease-modifying effects, we evaluated the proteomics and metabolomics from the somatosensory cortex and hippocampus of our TLE rat model. Our untargeted protein investigation found that phosphatase 2 regulatory subunit B α (Ppp2r5a), heat shock protein family A member 2 (Hspa2) and lymphocyte cytosolic protein 1 (Lcp1) were differentially expressed in the somatosensory cortex between selenate and vehicle-treated post-SE rats (Tukey’s HSD, FDR<0.05, Figure 3A, Supplementary File 1). We utilised semi-automated western blots to further validate the modification of the h-tau pathway by sodium selenate in chronically epileptic rats, and found that sodium selenate treatment persistently reduced h-tau compared to vehicle and levetiracetam treated animals (p <0.01 for both comparisons, Figure 3—figure supplement 3A). Similarly, PP2A protein expression was significantly increased in selenate-treated animals compared to vehicle (p <0.01), levetiracetam (p <0.01) and shams (p <0.05, Figure 3—figure supplement 3B).

Also novel is the analyses investigating the effect of sodium selenate treatment in chronically epileptic rats on telomere length. Telomere length has emerged as a potential prognostic biomarker of chronic neurodegenerative conditions (Chan et al., 2020, Kang et al., 2020, Galletly et al., 2017, Lukens et al., 2009). Our hypothesis was that selenate could be a modifier of neurodegeneration pathways, therefore we would anticipate that selenate treatment would prevent telomere shortening. We found that telomere length was significantly reduced in TLE animals treated with vehicle or levetiracetam, and that treatment with sodium selenate mitigated telomere length shortening. The magnitude of the reduced telomere length was strongly associated with the number of seizures the rats manifest, and more pronounced cognitive deficits, in our drug-resistant TLE model. Overall, our data highlights the potential of telomere length as a prognostic biomarker of disease severity and treatment response. Moreover, telomere length has the potential to be used in further trials of DMT (Pg 29-30).

However, unlike individual markers, pathway/network-based prognostic markers should be inherently more reliable since they provide the biological interpretation for the association between multiple molecules and the disease characteristics (seizures, behaviour comorbidities, development of TLE) (Zou et al., 2013) (Pg 30, lines 10-18). Therefore, in an attempt to shed light on the mechanisms by which sodium selenate is disease-modifying in established drug-resistant TLE, we performed a network medicine integration (Pg 27-29) of our untargeted/targeted multi-omics datasets, TLE outcomes (seizures and behaviour) and molecular studies (western blots and telomere length analyses), to evaluate the correlation coefficients of single modules between the different treatment groups. By mapping significant genes/proteins together with metabolites to biological pathways, our network integration approach allowed for a reduction in experimental and biological noise and higher confidence levels, thereby generating potential new disease modules (proteins and metabolites and the cellular pathways they are involved in).

We found modules that constitute the “healthy state”, which operate in a concerted fashion in a healthy brain but are lost in the setting of TLE. The Green Module is positively correlated to epilepsy phenotype, poor cognitive performance, poor balance, shorter telomeres, decreased PP2A, increased h-tau expression, and cardiomyopathies, which have been shown to be associated with several epilepsy syndromes. Of particular interest are the Yellow-Module in both the hippocampal and somatosensory cortex correlation networks as well as Tan and Light-cyan Modules in the somatosensory cortex. All of the above-mentioned modules positively correlated with selenate treatment. The Yellow Module is enriched for pathways involved in ATP metabolism, oxidative phosphorylation, and several inflammatory systems. The Tan Module is enriched for insulin signalling pathways and the Light-cyan Module annotates to fatty acid metabolism pathways. It is important to note that “enrichment” does not mean “increase” or “upregulation”. The enrichment in insulin signalling, for example, highlights the involvement of this pathway in the epilepsy phenotype and selenate treatment, but unfortunately provides no information about the direction of change (upregulation or downregulation). To capture changes in the regulation of biological pathways that may be influenced by sodium selenate treatment we conducted a quantitative enrichment analysis on a pathway level (PADOG) between the selenate and vehicle groups, utilizing the Reactome pathway database resource. Based on the PADOG analysis results, there is significant upregulation of IGFR1 and FGFR signalling cascades and FGFR-mediated PI-3K cascade activation in the selenate treated group compared to vehicle. Given the neuroprotective role of PI-3K signalling in several epilepsy syndromes, it is conceivable that this is one of the potential mechanisms by which sodium selenate exerts its long-term disease-modifying effect. The strong correlation of the Tan Module, which annotates to insulin signalling pathways to selenate treatment only further highlights the involvement of insulin signalling in the potential mechanism of action of this treatment. The downregulation of fatty acid oxidation processes in the selenate group suggests a shift from hyperactive neurons with high-energy demand (like in epileptic seizures) towards neurons with lower energy demand.

Our investigation supports our hypothesis about the importance of the molecules associated with the tau pathway and the potential value of telomere length as a surrogate marker of disease progression and response to treatment. However, future studies should incorporate multi-omics and telomere length analysis integration performed at different time points from brain tissue and biofluids (blood and CSF) to map out the development and progression of epilepsy and treatment response in this model (Pg 30). Further investigation of these pathways as potential mechanisms involved in the mode of action of sodium selenate can be instrumental in understanding and reversing the drug resistance of TLE. To our knowledge, this analysis is the first attempt of integration of metabolomic and proteomic data in an epilepsy network medicine paradigm.

Reviewer #1 (Recommendations for the authors):To improve the paper, the authors should consider how additional experiments could make it novel and a significant advance. Defining the mechanism would be an example. Additional endpoints typically studied or additional models might be useful if they shed light on the mechanism. Clinical relevance would be increased by human data.

Thanks for the insight, however, as described above, our current work is highly novel and clinically relevant, representing a major advance in the field being the first demonstration of a pharmacological disease-modifying therapy in a model of chronic, established epilepsy. Nevertheless, we have included ideas for future experiments to build up on our molecular findings and further dissect the mechanism of action of sodium selenate (Pg 29-30)

Reviewer #2 (Recommendations for the authors):The multi-omics analysis appears to be the weakest part of the manuscript. Given the interindividual variability and complexity of proteomic and metabolomic data, an n of 3 per group is small. In the Materials and methods section, the authors mention a power analysis that suggested a group size of 8. Please comment.

The power calculation we provided relates to the epilepsy/behavioural outcomes only and we have now clarified this in the manuscript. We acknowledge that 3 animals per group for the multi-omics analysis may be considered a limitation of our study, but unfortunately, these were the only tissue specimen available for this analysis. We believe that by combining data from two experimental modalities, using an early integration analysis approach, we optimise the opportunity to identify meaningful biological insights. We cite Schouten et al. work (Pg 47), as an example of the utility of a multi-omics approach using a similar sample size (Schouten et al., 2016),

The authors state that network medicine integration highlights discrete protein-metabolite modules that correlate with treatment and pre-clinical outcomes. This part needs much more attention and clarification. There is no individual discussion of the proteomic and metabolomic data including some supporting Figures (e.g., multivariate analysis/PCA, volcano plot, hierarchical clustering). Results from this section should not be "hidden" in the supplements.

We thank the reviewer for the comment. We decided to discuss only the multi-omics integration to facilitate the readability of the main manuscript. The discussion of the multivariate analysis/PCA, volcano plot, and hierarchical clustering as suggested by the reviewer, is interesting but perhaps is best reserved for a more technical paper. Our aim is to demonstrate that sodium selenate has a disease modifying effect in animals with established chronic epilepsy, and to use multi-omics network medicine and bioinformatics approaches to uncover relevant molecular pathways correlated to the response to selenate treatment and the epilepsy phenotype. However, we have now provided partial least squares-discriminant analysis (PLS-DA) and principal component analysis (PCA) plots for each proteomics metabolomics study from each brain region (Figure 3—figure supplement 2). The results of the univariate analysis on Pg 14-16, Figure 3 and we have now added the top 25 differentially expressed proteins as Figure 3—figure supplement 1. The rest of the omics data and all the data is included in the supplementary material or accessible through the public database Dryad.

Figure 5 is not intuitive. Please consider using different labels with a clear association to their gene/protein context, e.g, metabolism instead of the red module. Please provide more specific information on module size, gene content, and associations. A network graph (node size = module size, color code of nodes = Spearman correlation coefficient) may be used to help visualization. Are there any proteins or modules that distinguish treatment groups?

To address this concern, we have now labelled each module by its corresponding representative enriched pathway (Figure 5C and 5D, Pg 22-23). We thank the reviewer for the visualization suggestion, but feel this option would not work well due to the size of the modules identified in our analyses.

Figures 1-4: Please stick to a uniform color code across all figures, so that, e.g., Lev or Sodium selenate treated animals always have the same color in all graphs. The authors further may consider changing the labeling to support a more intuitive understanding of the figures, e.g., sham-control, post-SE+vehicle, post-SE+Lev, post-SE+Sodium selenate

Thanks for the comment, we have modified the figures, maintaining consistency and have changed the labelling as suggested by the reviewer.

[Editors' note: further revisions were suggested prior to acceptance, as described below.]

The manuscript has been improved but there are some remaining issues that need to be addressed, as outlined below:Recommendations:1. Remove all statements that PPAR, tau, telomeres, (etc.) and transcriptomics data caused effects of selenate. These data are only correlative.

We thank the reviewers and editors for their comments. We have removed and clarified the meaning of those statements throughout the manuscript as suggested by the reviewer. Changes have been incorporated on page 1-2; page 17 line 3; page 23 line 19-20, 24; page 24 line 1-2, page 25 lines 5-11, and page 29 lines 1-3, 18-19.

2. Revise statistics so that where variance is unequal in the different groups all data are either transformed (common methods are to take the log or reciprocal of all data points) to resolve the heterogeneity (making parametric statistics possible to use), or (when transformation does not resolve heterogeneity of variance) non-parametric statistics are used. To test whether variance is unequal, Bartlett's test and Brown-Forsythe are used.

We appreciate the feedback from the reviewers. We have reviewed all the statistics and modified the statistical analysis as requested. All EEG parameters and MWM were normally distributed as per the Shapiro-Wilk test. EEG data were compared using a two-way repeated measure ANOVA. Variance was assessed using the Brown-Forsythe test. For non-normally distributed data or where variance was unequal, a log10 transformation was used (i.e. SPT, NOP) to resolve the heterogeneity. Normally distributed, (OF, EPM, NOR, BT, TL and PP2A western blot) data were analyzed using one-way ANOVA with Tukey post hoc. The data that did not fit a normal or Gaussian distribution (NOP, NOR, SPT, h-tau/total tau western blot) was unequal after a log10 transformation (SPT) the non-parametric Kruskal-Wallis test with Dunn’s post hoc was used instead. Changes are reflected in the figures and the statistical analysis subsection (page 45-46).